# ReVision: Refining Video Diffusion with Explicit 3D Motion Modeling

**Qihao Liu**                                                          *qliu45@jhu.edu*
*Johns Hopkins University*

**Ju He**                                                              *jhe47@jhu.edu*
*Johns Hopkins University*

**Qihang Yu**                                                          *yucornetto@gmail.com*
*Independent Researcher*

**Liang-Chieh Chen**[†]                                               *lcchen@cs.ucla.edu*
*Independent Researcher*

**Alan Yuille**[†]                                                    *ayuille1@jhu.edu*
*Johns Hopkins University*

**Reviewed on OpenReview:** *https://openreview.net/forum?id=mQ5frFQTFV*

## Abstract

In recent years, video generation has seen significant advancements. However, challenges still persist in generating complex motions and interactions. To address these challenges, we introduce ReVision, a plug-and-play framework that explicitly integrates parameterized 3D model knowledge into a pretrained conditional video generation model, significantly enhancing its ability to generate high-quality videos with complex motion and interactions. Specifically, ReVision consists of three stages. First, a video diffusion model is used to generate a coarse video. Next, we extract a set of 2D and 3D features from the coarse video to construct a 3D object-centric representation, which is then refined by our proposed parameterized motion prior model to produce an accurate 3D motion sequence. Finally, this refined motion sequence is fed back into the same video diffusion model as additional conditioning, enabling the generation of motion-consistent videos, even in scenarios involving complex actions and interactions. We validate the effectiveness of our approach on Stable Video Diffusion, where ReVision significantly improves motion fidelity and coherence. Remarkably, with only 1.5B parameters, it even outperforms a state-of-the-art video generation model with over 13B parameters on complex video generation by a substantial margin. Our results suggest that, by incorporating 3D motion knowledge, even a relatively small video diffusion model can generate complex motions and interactions with greater realism and controllability, offering a promising solution for physically plausible video generation. Project page: https://revision-video.github.io/

## 1 Introduction

Video diffusion models have achieved remarkable success in producing high-quality, temporally coherent videos (Blattmann et al., 2023; Brooks et al., 2024; Polyak et al., 2024; Kong et al., 2024). It has been driven by advances in model architectures (Peebles & Xie, 2023), increases in model complexity, reaching tens of billion parameters (Polyak et al., 2024), and the availability of large-scale high-quality datasets (Chen et al., 2024). However, current models still struggle to generate videos that adhere to realistic physical

---

[†]Equal advising.

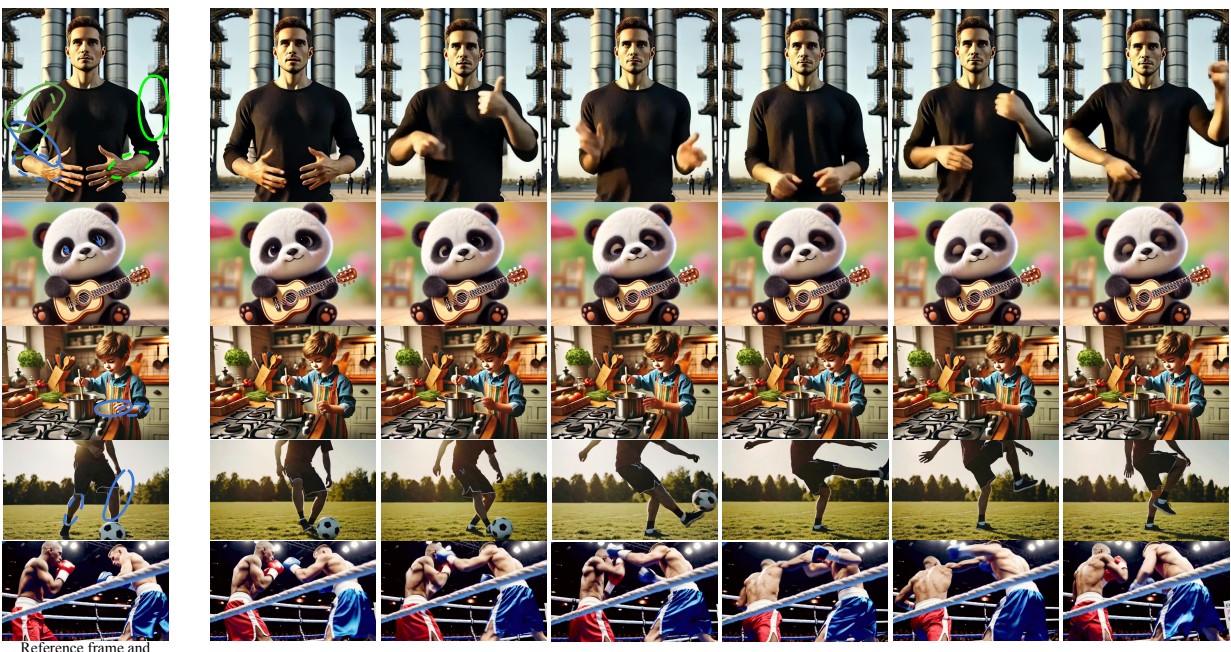

Reference frame and
user input target pose

Generated video sequence

Figure 1: By explicitly leveraging a parameterized 3D motion model, ReVision enhances pre-trained video generation models (*e.g.*, Stable Video Diffusion) to produce high-quality videos with complex motion (row 1), enabling precise motion control (rows 2, 3) and accurate interactions (rows 4, 5). During inference, an *optional* target pose can be specified via a rough sketch (rows 1, 3, 4, colored circles for different parts, dashed lines for the original pose, solid lines for the target pose) or a simple drag operation (blue arrows in row 2) indicating the final position.

principles, making it difficult to consistently achieve fine-grained motion control, complex movements, and coherent object interactions. Despite extensive efforts to improve performance through larger models and higher-quality datasets, a recent study (Kang et al., 2024) indicates that scaling model size and data alone is insufficient to fully capture the complexities of the real world.

On the other hand, human image animation models (Hu, 2024; Tan et al., 2024; Xu et al., 2024) offer valuable insights for addressing persistent challenges in video generation. Despite using smaller models and less data, these methods achieve consistent and precise video outputs with complex motions by following predefined 2D keypoint trajectories. This success suggests that incorporating a well-defined motion prior can substantially reduce the learning complexity of video generative models, enabling them to generate coherent and lifelike motion. However, in general video generation tasks, such strong guiding signals are typically unavailable, limiting the direct applicability of these animation techniques to broader video generation scenarios. This raises a critical question: *Can we develop a video generation framework that leverages the implicit motion information embedded in the generated videos as guidance to further enhance video quality?*

In this paper, we propose a simple, general, and plug-and-play video generation framework that incorporates motion knowledge into a conditional video generation model via a parameterized 3D representation, allowing the generation of videos with complex motions and interactions involving *humans*, *animals*, and *general objects*. The core of **ReVision** is to **Re**generate **Vi**deos with explicit 3D motion representat**ion**s, following an *Extract–Optimize–Reinforce* pipeline. Specifically, to effectively leverage 3D knowledge without heavy retraining of the diffusion model while preserving its original ability to generate high-quality visual appearance, we design the pipeline in three stages.

In the first stage, we employ a video diffusion model, *e.g.*, SVD (Blattmann et al., 2023), to generate a coarse video conditioned on the given input. In the second stage, we utilize parametric 3D models (*i.e.*, SMPL-X (Pavlakos et al., 2019) for humans, SMAL (Rueegg et al., 2023; Zuffi et al., 2024) for animals, and 2D binary mask (Yu et al., 2023) with estimated depth (Yang et al., 2024b) for general objects) to *extract* 3D

shape and motion features from the coarse video. These 3D object-centric representations are subsequently *optimized* by the proposed **P**arameterized **M**otion **P**rior model (**PMP**), producing a more accurate and natural 3D motion sequence. In the third stage, the refined 3D motion sequences are incorporated as additional conditioning inputs to *reinforce* the diffusion model, enabling it to regenerate the video with improved coherence and realism.

Extensive qualitative results and human preference studies confirm that our model excels at generating complex motions and interactions. We first apply ReVision on Stable Video Diffusion (SVD) (Blattmann et al., 2023), substantially improving its ability to generate realistic and intricate motions. We further compare ReVision-SVD with HunyuanVideo (Kong et al., 2024), a state-of-the-art open-source video generation model with 13B parameters, and demonstrate superior motion quality. Finally, on the particularly challenging dance generation task, our model outperforms state-of-the-art human image animation methods that rely on ground-truth pose sequences, surpassing them across all evaluation metrics.

In summary, we make the following contributions:

- We show that optimizing object-centric knowledge of generative models enhances their ability to generate complex motions and interactions, suggesting a promising direction for improving video generation.

- We introduce ReVision, a three-stage pipeline that significantly improves the quality of pre-trained video generation models by explicitly optimizing parameterized 3D object-centric motion knowledge extracted from generated videos.

- We propose PMP, a lightweight and robust parameterized motion prior model that effectively refines motion information in generated videos.

## 2   Related Work

**Video Generation.** With the success of diffusion models in image generation (Rombach et al., 2022; Esser et al., 2024; Liu et al., 2024a; Betker et al., 2023), driven by advancements in both generative modeling strategies (Ho et al., 2020; Song et al., 2020; Lipman et al., 2022; Liu et al., 2022; 2024b) and model architectures (Bao et al., 2023; Peebles & Xie, 2023; Liu et al., 2024c; Ma et al., 2024), video generation (Ho et al., 2022; Singer et al., 2022; Wang et al., 2024d; Yang et al., 2023; Zhang et al., 2024a; Zhou et al., 2022; Bar-Tal et al., 2024; Polyak et al., 2024; Brooks et al., 2024) has recently attracted significant attention. Parallel to text-to-video (T2V) generation, image-to-video (I2V) methods (Babaeizadeh et al., 2017; Li et al., 2018; Xiong et al., 2018; Pan et al., 2019; Zhang et al., 2020) generate videos from a single starting frame. However, existing methods still struggle to handle complex motions and interactions, and often fail to maintain physical plausibility. To overcome these challenges, recent approaches incorporate additional conditions to enhance motion control in video generation. Common conditional inputs include text descriptions (Hu et al., 2022; Girdhar et al., 2023; Chen et al., 2023; Ren et al., 2024b; Zeng et al., 2024), which can further guide motion modeling. For example, MAGE (Hu et al., 2022) introduces a spatially aligned motion anchor to blend motion cues from text, and SEINE (Chen et al., 2023) uses a random-mask video diffusion model to create transitions guided by textual descriptions. Another popular condition is optical flow, where models (Mahapatra & Kulkarni, 2022; Ni et al., 2023; Shi et al., 2024) estimate rough flow from user-provided arrows or text to guide complex motion generation. In contrast, ReVision leverages implicit motion features already embedded in the generated video through 3D parameterized object representations. This allows it to directly extract, optimize, and reinforce accurate and reliable motion features from the generated video itself, resulting in precise motion sequences that enhance coherence and fidelity.

**Human Image Animation.** Human image animation focuses on transferring motion from a source human to a target human by *using ground-truth posture sequences*, which can be represented as flow (Wang et al., 2004), keypoints (Hu, 2024; Tan et al., 2024), or human part masks (Xu et al., 2024). Extensive efforts have gone into extracting improved motion features. For example, MagicAnimate (Xu et al., 2024) leverages an off-the-shelf ControlNet (Zhang et al., 2023a) to obtain motion conditions, Hu *et al.* (Hu, 2024) introduce a Pose Guider network to align pose images with noise latents, and Animate-X (Tan et al., 2024) utilizes both implicit and explicit pose indicators to generate motion and pose features. Such strong guidance enables

high-quality video generation in human image animation, as each posture sequence directly dictates the synthesis of corresponding frames. However, *in general video generation, the ground-truth dense guidance is typically unavailable*, and there is usually no reference video for extracting a motion sequence. To overcome this limitation, ReVision introduces a three-stage process: it first extracts an implicit, rough motion sequence from the generated video, then refines it using the proposed PMP, and finally leverages the refined motion to guide video regeneration. This approach provides effective guidance for video generation, significantly improving video quality.

## 3 Preliminary

**Latent Diffusion Model.** Diffusion models (Ho et al., 2020) generate data through a denoising process that learns a probabilistic transformation. Latent diffusion models (Rombach et al., 2022) move this process from pixel space to the latent space of a Variational Autoencoder (Kingma & Welling, 2014). Specifically, we consider the latent representation $z_0$ of the input data. In the forward diffusion process, Gaussian noise is incrementally added to $z_0$:

$$q(z_t|z_{t-1}) = \mathcal{N}\left(z_t; \sqrt{1-\gamma_t}z_{t-1}, \gamma_t \mathbf{I}\right), \tag{1}$$

where $z_t$ represents the noisy latent representation at time step $t$, and $\gamma_t$ is a predefined noise schedule with $t \in (0, 1)$. As $t$ increases, the cumulative noise applied to $z_0$ intensifies, gradually transforming $z_t$ closer to pure Gaussian noise. We express the transformation from $z_0$ to $z_t$ directly as: $z_t = \sqrt{\bar{\alpha}_t}z_0 + \sqrt{1-\bar{\alpha}_t}\,\epsilon$, where $\bar{\alpha}_t = \prod_{i=1}^{t}(1-\gamma_i)$ and $\epsilon \sim \mathcal{N}(0, \mathbf{I})$. The latent diffusion model, parameterized by $\Theta$, learns to reverse this noising process by taking $z_t$ as input and reconstructing the clean data with the objective: $\mathcal{L} = \|\epsilon - \epsilon_\Theta(z_t, t, c)\|_2^2$, where $c$ is the condition to guide the denoising process. Once the latent space is reconstructed, it is decoded via the VAE decoder.

**Video Latent Diffusion Model.** We use SVD (Blattmann et al., 2023) as our base video diffusion model, which extends Stable Diffusion 2.1 (Rombach et al., 2022) to video. The main architectural difference from image diffusion models is that SVD incorporates a temporal UNet (Ronneberger et al., 2015) by adding temporal convolution and (cross-) attention (Vaswani, 2017) layers after each corresponding spatial layer.

**3D Human and Animal Mesh Recovery.** We utilize the SMPL-X (Pavlakos et al., 2019) and SMAL (Zuffi et al., 2017) parametric models to represent humans and animals, respectively. These models parameterize 3D meshes with pose parameters $\theta$ and shape parameters $\beta$. Additionally, SMPL-X model includes expression parameters $\psi$ to capture facial expressions through blend shapes. Given these parameters, SMPL-X model is a differentiable function that outputs a posed 3D human mesh $\mathcal{M}_{SMPL-X}(\theta_h, \beta_h, \psi_h) \in \mathbb{R}^{10475 \times 3}$, where pose $\theta_h \in \mathbb{R}^{165}$, shape $\beta_h \in \mathbb{R}^{10}$, and expression $\psi_h \in \mathbb{R}^{10}$. Similarly, SMAL model represents a posed 3D animal mesh with $\mathcal{M}_{SMAL}(\theta_a, \beta_a) \in \mathbb{R}^{3889 \times 3}$, where pose $\theta_a \in \mathbb{R}^{105}$ and shape $\beta_a \in \mathbb{R}^{41}$. In our work, we recover 3D human and animal meshes by fitting the SMPL-X and SMAL models to both our data and the generated videos. This produces 3D mesh reconstructions for all humans and animals. Because these meshes are computer-graphics models with predefined body-part annotations at every vertex, we can obtain accurate part labels directly. The 3D meshes also allow us to compute motion strength by measuring movement speed in 3D space, which is more reliable than relying on 2D pixel motion alone.

**2.5D Parameterized Object Representation.** Unlike humans and animals, there is no straightforward way to parameterize general objects in 3D space. Here, we represent objects in 2.5D by combining 2D bounding boxes (Varghese & Sambath, 2024), segmentation masks (Peng et al., 2020; Wu et al., 2024), and estimated depth (Yang et al., 2024b).

## 4 Method

ReVision requires extending a pre-trained video diffusion model to accept additional motion conditions as input. In Sec. 4.1, we describe how to adapt SVD into a motion-conditioned video generation model with minimal modifications. In Sec. 4.2, we introduce ReVision, a three-stage video generation pipeline built upon the extended SVD, incorporating a Parameterized Motion Prior model (PMP) to provide accurate motion sequences as conditioning inputs. An illustration is provided in Fig. 2.

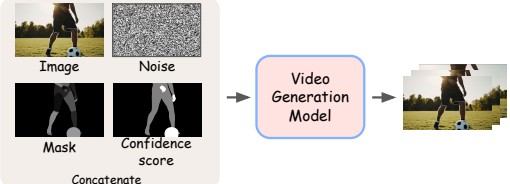

Figure 2: **Method overview.** Given the video generation model, ReVision operates in three stages. Stage 1: A coarse video is generated based on the provided conditions (*e.g.*, target pose, marked in blue, indicating the rough position of the yellow part in the last frame). Stage 2: 3D features from the generated coarse video are extracted and optimized using the proposed PMP. Stage 3: The optimized 3D sequences are used to regenerate the video with enhanced motion consistency. Best viewed when zoomed in.

## 4.1 Motion-Conditioned Video Generation

Since SVD does not natively support motion-conditioned video generation, we extend its design to enable this capability, with a focus on simplicity to preserve its original generation quality and minimize deviations from user-provided inputs. Concretely, we begin with a pre-trained SVD and fine-tune it within a carefully structured strategy. We concatenate two additional conditioning channels to the original condition: one for a part-level segmentation mask derived from the 3D motion sequence, and another for a confidence map indicating the reliability of the part mask, as illustrated in Fig. 3. We also design a fine-tuning pipeline that integrates three scenarios with varying levels of part mask guidance, allowing the model to flexibly handle diverse inputs. We detail those three scenarios below.

Figure 3: **Motion-conditioned video generation.** We enable motion-conditioned generation by introducing two extra conditioning channels: (1) part segmentation mask derived from the 3D motion sequence, and (2) its corresponding confidence map.

First, when the full motion sequence is provided (40% of training examples), the part-level mask is generated by merging all 2D part segmentation masks projected from 3D parametric mesh models. Since the motion sequence provides dense and precise control over video generation, we assign a confidence score of 1 to the corresponding confidence map. Our experiments confirm that these 3D-projected masks are more robust than existing part segmentation models.

Second, when only the target pose is provided (30% of training examples), we convert the projected part segmentation masks into polygons. This aligns with users' inference input, where they provide simple sketches (*e.g.*, circles or ovals) to indicate the final positions of specific targets or parts (*e.g.*, a hand or arm). These user-friendly sketches are then converted into polygonal masks, similar to the part segmentation mask polygons used during training. Since polygon conversion introduces unavoidable errors, we assign a confidence score of 0.5 in this case.

Last, to preserve SVD's ability to generate videos without motion conditioning, the remaining 30% of training examples provide an empty part mask, with a corresponding confidence score of 0.

Note that all three settings use the same model architecture, with minimal modifications limited to the first convolutional block of SVD. This design enables fine-tuning only the initial convolutional block and the temporal layers, avoiding the need to train SVD from scratch. As a result, the extended SVD can generate videos conditioned on various types of motion inputs, while still retaining its ability to generate videos from just text and the first frame.

## 4.2 Proposed Method: ReVision

**Overview.** As shown in Fig. 2, ReVision consists of three stages. In stage one (S1), we generate a coarse video based on the provided conditions. In stage two (S2), we extract both 2D and 3D features from the

coarse video and refine the 3D motion sequences through the proposed PMP. In stage three (S3), we use the refined 3D motion sequences as strong conditioning, guiding the video generation model to regenerate the video, resulting in significantly higher-quality output even for complex motions and interactions.

**S1: Coarse Video Generation.** Given the first frame and an *optional* user-specified target motion in the final frame, we use the fine-tuned SVD model to generate the video. Since the generation relies only on the target motion in the final frame or an empty motion, rather than a complete motion sequence, the resulting video often exhibits poor motion quality, leaving room for refinement. Therefore, we refer to this stage as coarse video generation.

Although we utilize only 2D and 3D motion features from the coarse video generated in Stage 1, this phase remains foundational, as it is critical for capturing rich motion patterns, intricate object interactions, and authentic camera movements in complex real-world settings. Video generation is inherently a multifaceted task. Beyond producing realistic object appearances, it also requires an understanding of dynamic motion, scene context, coherent camera trajectories, and the diverse interplay of these elements. While training a motion generation model directly for this task is theoretically feasible, it proves challenging in practice. Current state-of-the-art models (Guo et al., 2024; Zhang et al., 2024b; 2023b) are constrained by the lack of large datasets, limiting their capacity to model only simplistic motions, such as human-like activities like running or dancing. Consequently, they struggle to generalize to more complex motions, diverse object interactions, and fail to generate motions that align with realistic camera dynamics and scene context.

Stage 1 mitigates these limitations by directly generating videos and extracting detailed motion patterns, camera trajectories, scene transitions, and meaningful interactions. By leveraging extensive motion priors learned from billions of videos, it constructs a comprehensive sketch of motion and scene structure. This sketch serves as a critical foundation, providing the diversity and realism necessary to produce lifelike, engaging videos. The subsequent stages build on this: Stage 2 refines the motion further, while Stage 3 focuses on generating the video's visual appearance based on the refined motion feature.

Notice that, because only rough motion features without detailed visual information are needed from the coarse video, **we can significantly reduce computational overhead**. For instance, our experiments show that the compute time for Stage 1 can be reduced from 36 seconds to 8 seconds by generating the coarse video at a lower resolution (1/4 of the original), with fewer frames (1/2 of the original), and fewer denoising steps (32 vs. 50), while still preserving comparable final video quality. This optimization makes the overall generation process more efficient and cost-effective (See Tab. 3).

**S2: Object-Centric 3D Optimization.** After generating coarse videos, we parameterize the 3D information in the scenes for further optimization. For *humans* and *animals*, we employ well-established 3D parametric mesh models (Loper et al., 2015; Rueegg et al., 2023; Zuffi et al., 2024). For *general objects*, where no unified 3D representation or well-established modeling approach exists, we construct a parameterized representation by combining 2D bounding boxes (Varghese & Sambath, 2024), segmentation masks (Yu et al., 2023; Peng et al., 2020), and estimated depth (Yang et al., 2024b). Specifically, given the detected bounding box and segmentation mask, we extract a contour from the mask and approximate it with 16 vertices. We then combine these with 4 bounding box corners and the box center, yielding a total of 21 key 2D points. These points are lifted into 3D space using the estimated depth, resulting in a compact point-based representation for each object, denoted as $p_o \in \mathbb{R}^{21 \times 3}$.

However, due to the poor motion quality and inconsistencies in the coarse video generated in S1, the 3D parameters extracted also suffer from instability and inconsistencies. To address this, we propose a Parameterized Motion Prior model (PMP) to optimize the 3D motion sequence, based on the text information and the motion strength.

PMP first extracts text embeddings from the text description using a pre-trained CLIP encoder (Radford et al., 2021). Motion strength is computed from the differences in parametric 3D model parameters between adjacent frames, providing a measure of motion speed. Since the 3D motion sequences are already parameterized as 3D vectors, PMP employs a series of transformer blocks to iteratively refine the motion sequence based on these conditioning inputs. Within each block, motion features undergo self-attention, followed by cross-attention with the conditioning inputs (text embeddings and motion strength) and a feed-

forward network to generate the final output. Finally, the optimized 3D parameterized motion sequences are converted into 3D mesh sequences and projected into 2D as part segmentation masks and confidence maps, providing more accurate motion guidance. Architectural details are provided in Sec A of the Appendix, and the effectiveness of PMP is demonstrated in Sec. 5.3.

To train PMP, we introduce small perturbations to the ground-truth motion sequences from the annotated Panda-70M subset. Three types of perturbations are randomly applied: (1) adding small random noise to the motion sequence, (2) shuffling the internal order of the sequence, and (3) dropping a small segment while repeating the remaining segments to maintain the original length. Through this process, PMP learns to denoise perturbations, improving its ability to recover smooth and robust motion sequences.

**S3: Fine-grained Video Generation.** In the final stage, we regenerate the video using the same SVD model but with the improved motion sequence as additional conditioning. Unlike the coarse generation in stage one, which uses only the target pose in the last frame or no motion information, we now utilize the full motion sequence as part masks optimized in 3D space. With this stronger conditioning, the final output exhibits significantly improved motion consistency compared to the coarse video, as illustrated in Fig. 8.

# 5 Experimental Results

We first compare our method with SVD (Blattmann et al., 2023) and HunyuanVideo (Kong et al., 2024) in Sec. 5.1, highlighting how it enhances SVD to support more controllable and complex motion generation while maintaining efficiency, effectively handling occlusions, and enabling long video generation. Next, in Sec. 5.2, we compare our model with Human Image Animation methods, demonstrating its ability to generate complex motions. We then evaluate the effectiveness of the proposed Parameterized Motion Prior in Sec. 5.3. Due to space limitations, additional details and results, including ablations on parametric 3D mesh, text prompt, and motion strength, are provided in Sec. C of the Appendix.

## 5.1 Image-to-Video Generation

**Dataset.** Both the motion-conditioned video generation model and the Parameterized Motion Prior model (PMP) need to be fine-tuned (trained) on a small yet high-quality video dataset with object-centric annotations. Existing large-scale video datasets (Bain et al., 2021; Chen et al., 2024) mainly provide text-image pairs without detailed object-centric annotations. To address this limitation, we use a suite of off-the-shelf models across various tasks to generate 2D and 3D object-centric annotations. We annotate a total of 20K videos from the Panda-70M (Chen et al., 2024) dataset. For each video, we provide frame-wise 2D bounding boxes, semantic masks, depth estimation maps, and 3D parametric mesh reconstructions for detected humans and animals. The details are outlined in Sec. B in the Appendix.

**Experimental Setup.** For most experiments, we use SVD (Blattmann et al., 2023) as the base video generation model and modify it by introducing two additional channels for conditional generation. We fine-tune SVD on our annotated dataset for 300K iterations with a batch size of 64 and a constant learning rate of $2 \times 10^{-5}$. During training, we randomly sample 16-frame video clips with a stride of 4 at a resolution of $1024 \times 576$. To enable various control, we incorporate different conditioning strategies: 40% of video clips contain accurate part masks for each frame, 30% contain a polygon mask for random parts in the final frame, and the remaining clips have no additional conditioning.

**Benchmark on General Video Generation.** To better evaluate our model on general video generation, we used VBench++ (Huang et al., 2024), which provides a comprehensive, detailed, multi-dimensional assessment of general video generation quality. As our model is designed for image-to-video generation (I2V), we primarily compared it against SVD-XT-1.1 (Blattmann et al., 2023). Moreover, since our model is backbone-agnostic, we integrate our method with a stronger and more recent video diffusion backbone, Wan2.1-I2V-14B-720P (Wan et al., 2025). We train ReVision-Wan2.1 on the same dataset as ReVision-SVD using the same settings. Results are provided in Tab. 1. Our models consistently outperforms the base models on nearly all metrics, particularly in dynamic degree (83.15% vs. 43.17%, 73.67% vs. 51.38%) and various consistency and smoothness measures.

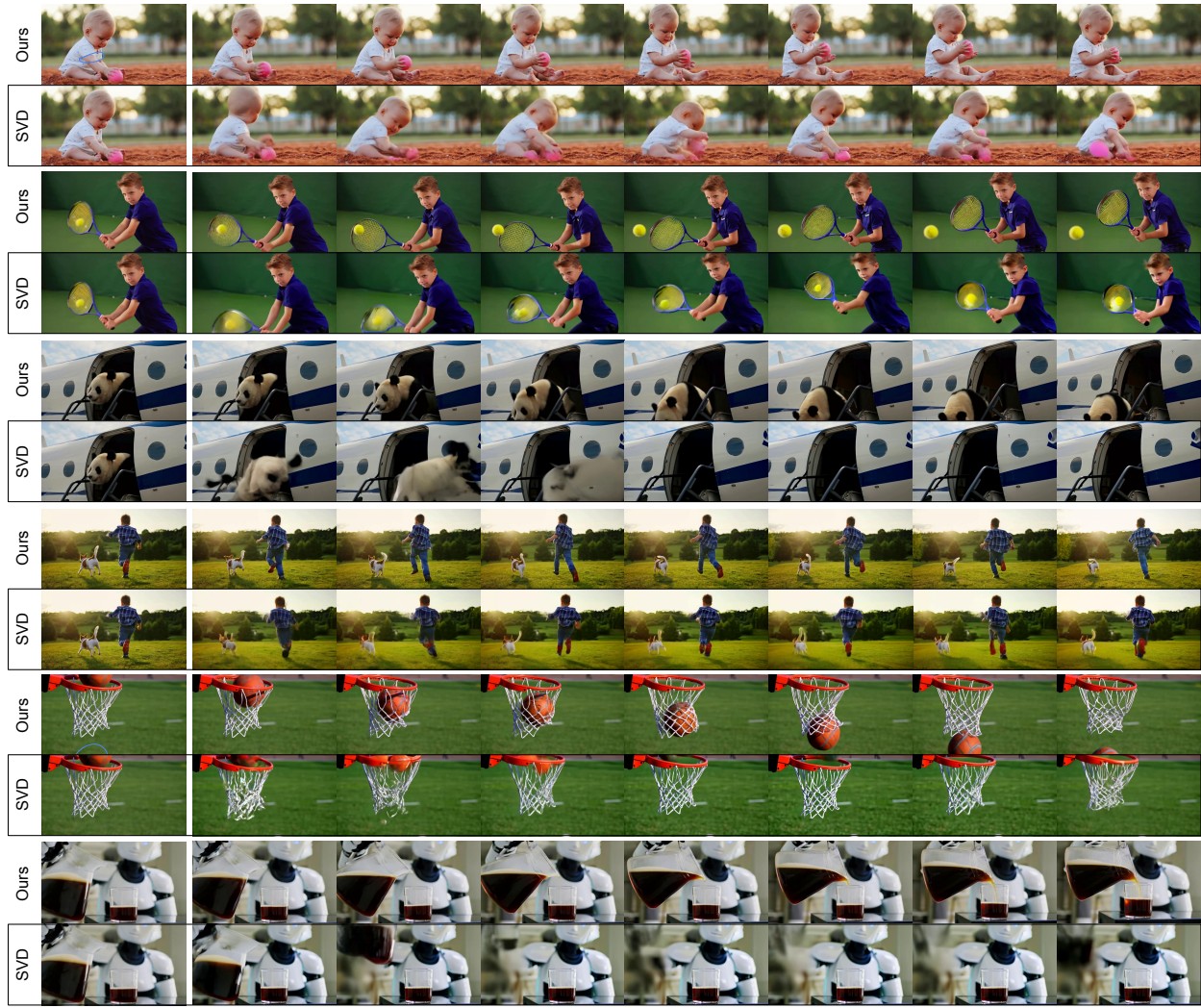

Figure 4: **Qualitative comparisons.** ReVision generates high-quality videos with complex motions and interactions of *humans, animals, and general objects.* Zoom in for better details. Please find the side-by-side video comparisons in the supplementary video. Reference frames are in the first column.

Table 1: **Quantitative comparisons on VBench++.** We achieve a significantly higher Dynamic Degree while maintaining similar performance across all metrics of consistency, smoothness, and quality.

| | Model Type | I2V Subject | I2V Background | Subject Consistency | Background Consistency | Motion Smoothness | Dynamic Degree | Imaging Quality |
|---|---|---|---|---|---|---|---|---|
| Step-Video-TI2V (Huang et al., 2025) | TI2V | 98.63% | 98.63% | 96.02% | 97.06% | 99.24% | 48.78% | 70.44% |
| DynamiCrafter-1024 (Xing et al., 2024) | TI2V | 98.17% | 98.60% | 95.69% | 97.38% | 97.38% | 47.40% | 69.34% |
| Gen-4-I2V (Runway, 2026) | I2V | 97.84% | 97.46% | 93.23% | 96.79% | 98.99% | 55.20% | 70.41% |
| Magi-1 (Teng et al., 2025) | I2V | 98.39% | 99.00% | 93.96% | 96.74% | 98.68% | 68.21% | 69.71% |
| HunyuanVideo-I2V (Kong et al., 2024) | I2V | 98.53% | 97.37% | 95.26% | 96.70% | 99.23% | 22.20% | 70.1% |
| Wan2.1-I2V-14B-720P (Wan et al., 2025) | I2V | 96.95% | 96.44% | 94.86% | 97.07% | **97.90%** | 51.38% | 70.44% |
| **ReVision-Wan2.1 (Ours)** | I2V | **98.10%** | **97.10%** | **97.06%** | **97.89%** | 97.74% | **73.67%** | **72.86%** |
| SVD-XT-1.1 (Blattmann et al., 2023) | I2V | 97.51% | 97.62% | 95.42% | 96.77% | 98.12% | 43.17% | 70.23% |
| **ReVision-SVD (Ours)** | I2V | **97.94%** | **98.06%** | **96.13%** | **97.89%** | **98.88%** | **83.15%** | **71.48%** |

In addition, we compared our method with recent state-of-the-art models for text+image-to-video (TI2V) generation in Tab. 1. We observe that, although these TI2V models use text descriptions to specify motion, the intensity and quality of the resulting movements are not fully controllable. In contrast, our method

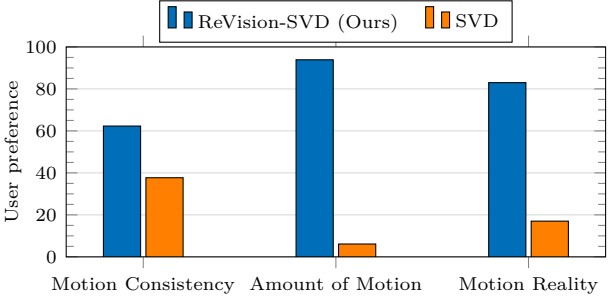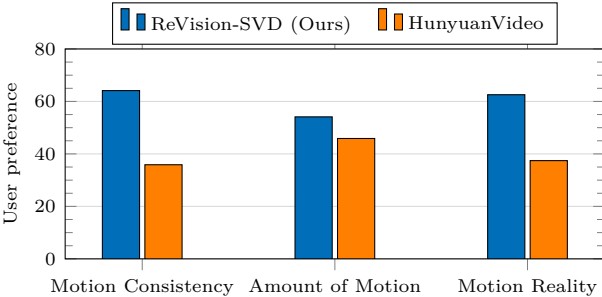

Figure 5: **User preference comparisons.** Our model enhances the motion generation capability of the pre-trained SVD. It even surpasses HunyuanVideo, a SOTA model with 13B parameters. These results highlight the effectiveness of our model in generating complex motions and interactions.

Table 2: **Comparisons with motion-conditioned video generation methods on DAVIS.** ReVision achieves more accurate motion transfer while preserving object appearance and scene details, resulting in enhanced temporal coherence and reduced visual artifacts.

|  | CoTracker mIoU ↑ | Optical Flow Error ↓ | Pixel MSE ↓ | Subject Consistency ↑ | Background Consistency ↑ | Motion Smooth ↑ |
|---|---|---|---|---|---|---|
| MotionClone (Ling et al., 2024) | 0.72 | 0.42 | 0.068 | 0.75 | 0.85 | 0.92 |
| ImageConductor (Li et al., 2025) | 0.66 | 0.64 | 0.072 | 0.77 | 0.88 | 0.93 |
| Go-with-the-Flow (Burgert et al., 2025) | 0.74 | 0.36 | 0.053 | 0.88 | 0.92 | 0.98 |
| **ReVision-SVD (Ours)** | **0.80** | **0.33** | **0.046** | **0.96** | **0.97** | **0.99** |

achieves substantially better performance in generating complex motions, while maintaining high consistency, temporal smoothness, and overall visual quality.

**Comparisons with Motion-conditioned Video Generation.** Following Go-with-the-Flow (Burgert et al., 2025), we also consider the motion-transfer image-to-video task on DAVIS (Pont-Tuset et al., 2017) and report the results in Tab. 2. We observe that our model consistently outperforms all baselines across all metrics, clearly demonstrating the effectiveness of the proposed method. In particular, our approach yields more accurate motion transfer while better preserving object appearance and scene details, leading to improved temporal coherence and fewer visual artifacts.

In addition, comparing text-conditioned and motion-conditioned video generation shows that explicit motion conditioning plays a dominant role in controlling dynamics, while text conditioning provides only coarse guidance. Although text descriptions effectively convey high-level semantic intent, they struggle to specify fine-grained temporal details and precise motion patterns. As a result, *motion conditioning enables more direct and reliable control over dynamic behaviors that are difficult to express through natural language.*

**User Study.** To better compare our model with the baseline SVD and HunyuanVideo, we conduct user studies to assess user preferences. Specifically, we generate 500 text descriptions of humans and animals engaged in daily activities using GPT-4o (Hurst et al., 2024). For the comparison with SVD, we use GPT-4o to generate five 16 : 9 images for each prompt, which are resized to $1024 \times 576$ as input. For the comparison with HunyuanVideo, we first use their released model to generate five videos at a resolution of $1280 \times 720$ for each prompt, then extract the first frame of each video as the input image for our model to generate the corresponding video. No target pose is provided for any model. For each image, we generate one video per model using the same random seed (42), resulting in a total of $5,000$ video pairs. Each video pair is evaluated by three randomly selected users on Amazon MTurk, leading to a total of $15,000$ comparisons. Users are shown two videos side by side, generated by different models, with the order randomized. They are instructed to assess the videos based on Motion Consistency, Amount of Motion, and Motion Realism. The results are reported in Fig. 5. Our model significantly enhances the motion generation capabilities of SVD, producing videos with superior motion quantity, consistency, and realism. Furthermore, it even surpasses HunyuanVideo, a state-of-the-art video generation model with 13B parameters, in terms of motion quality. These results highlight the effectiveness of our model in generating complex motions and interactions.

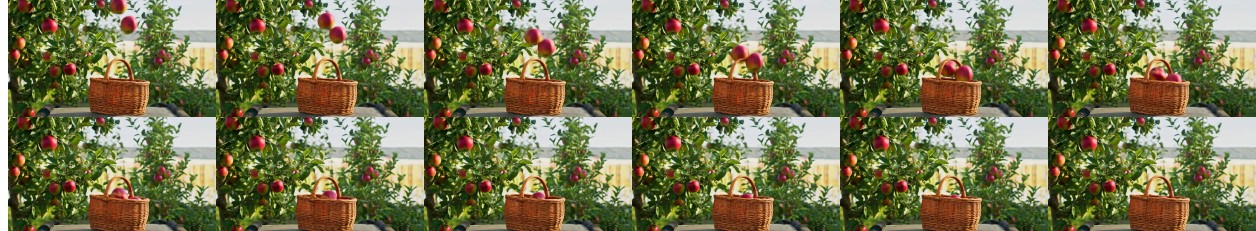

Figure 6: **Handling occlusion.** As illustrated by the two apples falling into the basket, ReVision handles occlusions by lifting and optimizing motion in 3D space, which allows explicit reasoning about object spatial relationships, effectively resolving occlusions that are ambiguous in 2D.

**Qualitative Comparisons.** We provide samples of generated videos in Fig. 4. Our ReVision produces realistic movements that closely follow user instructions. It also generates high-quality videos that involves complex motions and interactions, such as running with dogs, picking up a ball, and hitting a tennis ball. More visualizations are available in the supplementary videos.

**Inference Speed.** We compare the inference speed of our model against two baselines in Tab. 3. Despite the three-step pipeline, the coarse video generation (S1) takes only 8 seconds after our optimization, and the additional 3D detection and refinement modules (S2) add 5 seconds to the inference time on a single A100. Together, these two stages are significantly faster than the original SVD, which requires 36 seconds. More importantly, with just 1.5B parameters and a runtime of 49 seconds, our model generates high-quality videos with complex

Table 3: **Inference speed.** Average time to generate a 32-frame video. Our ReVision-SVD matches SVD in speed (8.4x faster than HunyuanVideo) while surpassing HunyuanVideo in generating complex motions and interactions.

|  | SVD | ReVision-SVD | HunyuanVideo |
|---|---|---|---|
| Time (s) | 36 | 49 (8 + 5 + 36) | 411 |

motions – comparable to or even surpassing state-of-the-art models like HunyuanVideo (see Fig. 4 and 5), which uses over 13B parameters and requires an average of 411 seconds.

**Handling Occlusion.** Occlusion becomes a significant challenge when generating videos with multiple objects and large motions. However, by lifting everything into 3D, occlusion is naturally resolved: Since we estimate depth, all objects are fully represented with their spatial positions, allowing us to reason about their relative locations in 3D. And after optimizing the motion in 3D, we project it back to 2D using the depth information, which restores accurate occlusion relationships in the camera coordinate. An example is shown in Fig. 6, where two apples are generated dropping into a basket. Our model effectively captures spatial relationships, producing realistic videos in which the apples fall *into* the basket with appropriate occlusion.

To further evaluate our model's ability to handle occlusion, we conducted an additional user study using the 5,000 video pairs generated for the main experiment. Users on Amazon MTurk were asked to assess the quality of occlusion handling and object interactions for each video pair. If no object interaction was observed, users were instructed to select "no interaction/occlusion." Similarly, each video pair was evaluated independently by three randomly selected users to ensure relia-

Table 4: **User preference comparisons for occlusion and interaction handling.**

|  | ReVision-SVD (Ours) | SVD |
|---|---|---|
| Preference | 97.63% | 2.37% |

|  | ReVision-SVD (Ours) | HunyuanVideo |
|---|---|---|
| Preference | 63.99% | 36.01% |

bility. The study yielded 15,000 evaluations, including 10,207 valid comparisons and 4,793 responses marked as "no interaction/occlusion". The results of these 10,207 valid comparisons are summarized in Tab. 4, highlighting that our model consistently outperforms both SVD and HunyuanVideo across a diverse range of occlusion scenarios.

**Long Video Generation.** Another advantage of our model is its ability to generate complex and large-scale motions over long video sequences (Fig. 7). PMP optimizes motion in a 3D parameterized space, enabling smooth and realistic interpolation and extrapolation to arbitrary lengths. The resulting long 3D motion sequences are then used to generate multiple overlapping video clips, which are stitched together to form extended videos with consistent motion. More specifically, we resample in motion-parameter space (not

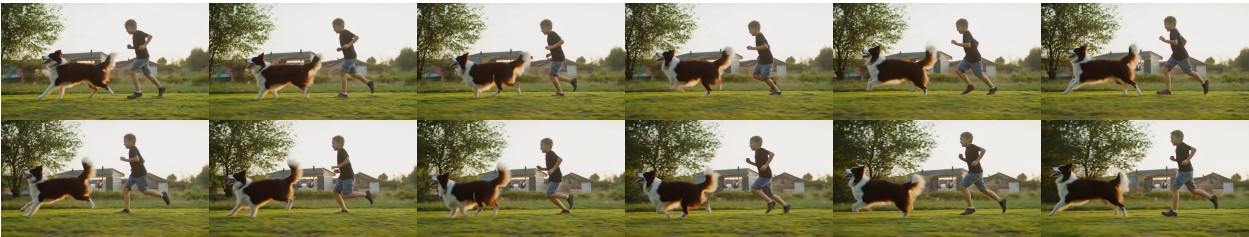

Figure 7: **Long video generation.** Our PMP extends a 32-frame 3D motion to 128 frames through interpolation (32 → 64), extrapolation (64 → 128), and refinement, enabling complex, large-scale motion generation over long video sequences. See supplementary videos for details.

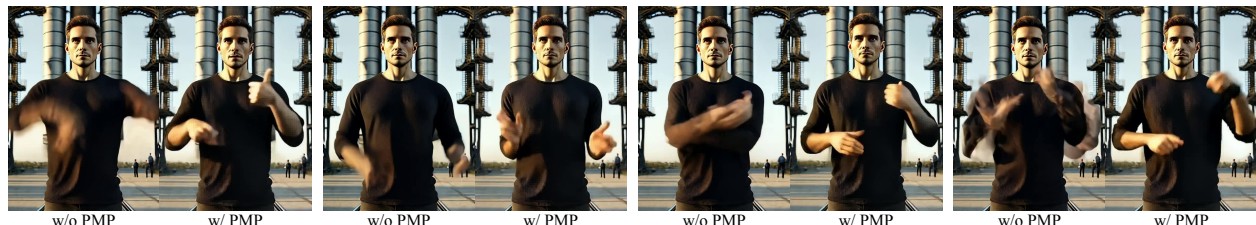

Figure 8: PMP improves motion and visual quality, generating realistic videos with large motions.

pixel space) through interpolation and extrapolation. Then, for long-video synthesis we generate overlapping 32-frame clips with a sliding window (stride 24, overlap 8) and stitch overlaps with a ramped blend in the overlapping frames. Motion conditions are aligned in the overlap and all windows share the same appearance/identity conditioning. *The 3D representation plays a key role in maintaining smooth temporal continuity.* While long video generation is not our main focus, using advanced techniques like temporal compression (Bar-Tal et al., 2024), beyond simple overlap, could further improve visual coherence. We leave their integration with our 3D-aware framework as future work.

## 5.2 Complex Motion Generation

**Experimental Setup.** To demonstrate our model's ability to generate videos with complex motion, we compare our approach with state-of-the-art human image animation models on the TikTok Dancing dataset (Jafarian & Park, 2021), using the Disco (Wang et al., 2024c) split. For compatibility with the SVD model architecture, all videos are cropped to $576 \times 1024$. We fine-tune the original SVD only on the training split for 30K iterations, with a batch size of 8 and a learning rate of $1 \times 10^{-5}$.

**Evaluation Metrics.** We follow baselines and report Peak Signal-to-Noise Ratio (PSNR) (Hore & Ziou, 2010), Structural Similarity Index (SSIM) (Wang et al., 2004), and Learned Perceptual Image Patch Similarity (LPIPS) (Zhang et al., 2018) to measure the visual quality of the generated results. We also report and Fréchet Video Distance (FVD) (Unterthiner et al., 2018) for video fidelity comparision.

**Experimental Results.** We compare ReVision with human image animation methods in Tab. 5, where we achieve state-of-the-art performance across all metrics. Notably, we observe a significant improvement in FVD, highlighting substantial gains in video generation quality. It is important to note that all baselines in this task *rely on ground-truth motion sequences*, which are challenging to obtain in practical scenarios, limiting their applicability. In contrast, our method can generate realistic and high-quality videos *using only the input inference image or inference image with a target pose.*

## 5.3 Parameterized Motion Prior Model (PMP)

We demonstrate the effectiveness of the proposed Parameterized Motion Prior model in this section.

Table 5: **Quantitative comparisons for dance generation.** 'ReVision (w/ full motion)' follows baselines and takes full motion sequences as condition, while 'ReVision (w/ target pose)' uses the target pose from the final frame.

| | SSIM ↑ | PSNR ↑ | LPIPS ↓ | FVD ↓ |
|---|---|---|---|---|
| MagicAnimate (Xu et al., 2024) | 0.714 | 29.16 | 0.239 | 179.07 |
| Animate Anyone (Hu, 2024) | 0.718 | 29.56 | 0.285 | 171.90 |
| Champ (Zhu et al., 2024) | 0.802 | 29.91 | 0.234 | 160.82 |
| VividPose (Wang et al., 2024b) | 0.758 | 29.83 | 0.261 | 152.97 |
| ReVision (image only) | - | - | - | 136.43 |
| ReVision (w/ target pose) | - | - | - | 130.14 |
| ReVision (w/ full motion) | **0.864** | **30.08** | **0.210** | **121.26** |

Table 6: **PMP acts as a general motion denoiser, improving performance on human motion generation.** Following MoMask, we report R-Precision at Top-1, Top-2, and Top-3. Our PMP achieves state-of-the-art performance on two widely used benchmarks.

| | HumanML3D | | | KIT-ML | | |
|---|---|---|---|---|---|---|
| | R-P@1 | R-P@2 | R-P@3 | R-P@1 | R-P@2 | R-P@3 |
| MotionDiffuse (Zhang et al., 2024b) | 0.491 | 0.681 | 0.782 | 0.417 | 0.621 | 0.739 |
| ReMoDiffuse (Zhang et al., 2023b) | 0.510 | 0.698 | 0.795 | 0.427 | 0.641 | 0.765 |
| MoMask (Guo et al., 2024) | 0.521 | 0.713 | 0.807 | 0.433 | 0.656 | 0.781 |
| MoMask + PMP | **0.544** | **0.735** | **0.810** | **0.471** | **0.673** | **0.785** |

**PMP Enables High-Quality Video Generation with Complex Motions and Interactions.** To demonstrate the effectiveness of PMP, we select a complex dance scenario and visualize outputs with and without the proposed PMP in Fig. 8. We also show quantitative improvements of the generated videos with PMP in Tab. 7, where 500 video pairs were evaluated by random users on Amazon MTurk. Each pair was rated by three different users, resulting in a total of 1,500 evaluations. The results show that the video generation model alone still struggles to produce high-quality videos with accurate motion. However, *leveraging the object-level priors from our Parameterized Motion Prior enables the generation of realistic videos with enhanced motion and visual quality.*

Table 7: **User studies for PMP.** PMP improves object and motion consistency, while reducing morphological failure rates.

| | Object Consistency ↑ | Motion Consistency ↑ | Morphological Failure Rate ↓ |
|---|---|---|---|
| w/o PMP | 12.4 | 4.0 | 83.5 |
| w/ PMP | 87.6 | 96.0 | 14.3 |

**PMP Prevents Error Accumulation in Multi-stage Video Generation.** In addition, Fig. 8 shows that even when the generated videos exhibit severely broken motion, our PMP can still recover (predict) a smooth and coherent motion sequence using the ground-truth first frame and target pose, enabling successful final video generation. *This correction mitigates motion errors and prevents error accumulation, highlighting the robustness of our pipeline.*

**PMP as a General Motion Denoiser.** We focus on a more specific human motion generation task and show that our model improves the performance of the state-of-the-art method, MoMask (Guo et al., 2024), on standard benchmarks (see Tab. 6). Specifically, we use PMP to refine the motion sequences generated by MoMask and compare the results with MoMask and other methods on HumanML3D (Guo et al., 2022) and KIT-ML (Plappert et al., 2016) benchmarks. We adopt the same training dataset as MoMask and apply the perturbations described in Sec. 4.2 to train PMP. *Serving as a general motion denoiser, our model consistently enhances motion generation quality of the current best models across multiple benchmarks.*

# 6 Conclusion

We introduced ReVision, a three-stage framework for video generation that improves motion consistency by integrating 3D motion cues. ReVision leverages a pretrained video diffusion model to generate coarse videos, refines 3D motion sequences with PMP, and reconditions the generation process with enhanced motions to improve fine-grained and complex motion generations. Evaluations show that ReVision significantly outperforms existing methods in motion fidelity and coherence.

**Acknowledgements.** Thanks to all who supported this project and to the anonymous reviewers for their constructive comments. AY acknowledges support from the ONR N000142412696.

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

## Appendix

The supplementary material includes the following additional information.

- Sec. A provides the architectural details for PMP.

- Sec. B provides additional details for our annotated dataset.

- Sec. C provides additional ablation studies omitted from the main paper.

- Sec. D discusses the limitations of our method.

- Sec. E discusses the societal impacts of our method.

We also provide the generated videos used in all figures in the main paper, as well as additional videos demonstrating accurate motion control, in the *supplementary videos.*

## A  Architectural details for PMP

To optimize the 3D motion sequence extracted from the coarse generated video, we propose the Parameterized Motion Prior model (PMP). As shown in Fig. 9, PMP utilizes a transformer architecture with self-attention, cross-attention, and feedforward layers as its backbone. It takes the parameterized motion sequence of the coarse video as input and optimizes it based on the input text prompt and motion strength. More specifically, PMP is trained as a single shared model across all categories. It directly predicts the corrected pose from the perturbed input, and we supervise it with an MSE loss between the predicted pose and the ground-truth pose.

## B  Dataset

Both the motion-conditioned video generation model and the Parameterized Motion Prior model (PMP) need to be fine-tuned (trained) on a small yet high-quality video dataset with object-centric annotations. Existing large-scale video datasets (Bain et al., 2021; Chen et al., 2024) mainly provide text-image pairs without detailed object-

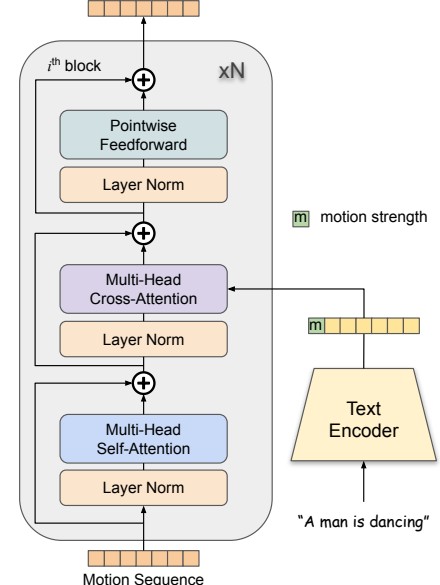

Figure 9: **Architecture of the Parameterized Motion Prior model**

centric annotations. To address this limitation, we use a suite of off-the-shelf models across various tasks to generate detailed 2D and 3D object-centric annotations. We annotate a total of 20K videos from the Panda-70M (Chen et al., 2024) dataset, with approximately 55% human videos, 15% animal videos, and 30% general-object videos. For each video, we provide frame-wise 2D bounding boxes, semantic masks, depth estimation maps, and 3D parametric mesh reconstructions for detected humans and animals. The details are outlined below.

**High-Quality Motion Videos Filtering.** To start with, we use LLMs (Yang et al., 2024a) and an open-vocabulary segmentation model (Yu et al., 2023) to curate high-quality motion videos. Specifically, LLM filters videos with evident motion based on their captions. Then, for each selected video, we equally sample 10 frames and apply the segmentation model to identify humans and animals. We evaluate each frame based on the predicted mask size and mask count. Then we retain videos where humans or animals occupy a significant portion of the frame and where the count of humans does not exceed five in each frame.

Table 8: **Quantitative evaluation regarding inference efficiency.** We reduce Stage 1 compute time from 36 seconds to 8 seconds, while maintaining comparable final video quality.

| | Overall Preference | Visual Quality | Motion Consistency | Amount of Motion |
|---|---|---|---|---|
| ReVision-SVD (full model) | 52.60% | 50.27% | 51.87% | 54.27% |
| ReVision-SVD (efficient model) | 47.40% | 49.73% | 48.13% | 45.73% |

**Object Detection and Depth Estimation.** Based on the captions of videos, we identify the objects mentioned and detect their bounding boxes (Varghese & Sambath, 2024) and instance masks (Yu et al., 2023). We also apply Depth Anything V2 (Yang et al., 2024b) to generate the depth maps of each frame.

**Human Videos Annotation.** For videos containing humans, we focus on extracting 2D instance segmentation masks, 2D part masks, 2D face keypoints, 3D body pose and shape, and 3D hand pose. We begin by using YOLO-V8 (Varghese & Sambath, 2024) to segment all humans in each frame, providing accurate human masks. Next, we apply a state-of-the-art face keypoint detector, RTMPose (Jiang et al., 2023), to predict facial keypoints for each detected human. Simultaneously, we use 4D-Human (Goel et al., 2023) and HaMeR (Pavlakos et al., 2024) to estimate the 3D body and hand meshes. The resulting SMPL (body mesh) (Loper et al., 2015) and MANO (hand mesh) (Romero et al., 2022) parameters are then fit into a unified SMPL-X (Pavlakos et al., 2019) representation, which contains both human body and hand meshes. We then project the 3D SMPL-X human mesh onto 2D to obtain part masks, as each vertex in the SMPL-X mesh is labeled by body part. Finally, we project the face keypoints and 3D human mesh onto the instance mask, allowing us to compute the overlap between the projected keypoints, projected human mask, and detected 2D human mask. This overlap is quantified using an IoU score, which is used to filter out annotations with high errors. As a result, for each video, we obtain annotations including human instance masks, 2D facial keypoints, 3D SMPL-X meshes for the body and hands, and 2D part-level segmentation masks.

**Animal Videos Annotation.** We start by using Grounded SAM 2 (Kirillov et al., 2023; Ravi et al., 2024; Ren et al., 2024a) to segment animal masks in each frame. Next, we apply a state-of-the-art camera estimation algorithm, VGGSfM (Wang et al., 2024a), to optimize the camera's intrinsic and extrinsic parameters across the video. To ensure a reliable camera estimate, we set thresholds on mean projection errors and mean track lengths, filtering out videos that do not meet these criteria. We then use AnimalAvatar (Sabathier et al., 2024) initialized with Animal3D (Xu et al., 2023) to fit SMAL parameters. Each video is divided into segments of 10 consecutive frames, and AnimalAvatar is applied to each segment independently. This strategy helps mitigate the impact of outliers in camera predictions on the overall optimization quality. To ensure the accuracy of SMAL fitting, we impose thresholds on IoU and PSNR (Hore & Ziou, 2010), filtering out video segments that do not meet our accuracy standards. Once accurate SMAL fittings are obtained, we follow a similar pipeline to extract the desired annotations as used in human cases.

## C  Ablation Study

Herein, we conduct additional ablation studies to verify the effectiveness of the proposed designs.

**Quantitative Evaluation Regarding Inference Efficiency.** Stage 1 can be performed with reduced computational overhead while maintaining similar results. To quantify this, we conducted experiments on 500 video pairs using the same input images but with varying settings: lower resolution (1:4), fewer frames (8 vs. 16), and fewer denoising steps (32 vs. 50). The efficient model reduced Stage 1 compute time from 36 seconds to 8 seconds. We ran a human preference study on Amazon MTurk, comparing the full and efficient versions for each video, with results provided in Tab. 8. The results demonstrate that the efficient model achieves similar performance to the full model in visual quality and motion consistency, with only a slight drop in the amount of motion, which is likely due to the reduced number of frames. This confirms that Stage 1 can be significantly accelerated with minimal impact on perceived quality.

**Parameterized Motion Prior Model.** We briefly discuss the quantitative improvements of the proposed PMP in the main paper and provide additional experimental details and results here. To further demonstrate the improvements of PMP, we conducted an additional user study on Amazon MTurk comparing videos generated with and without PMP. Unlike our previous study, which compared our method with SVD, this evaluation focuses on object consistency and motion consistency. We also report the percentage of

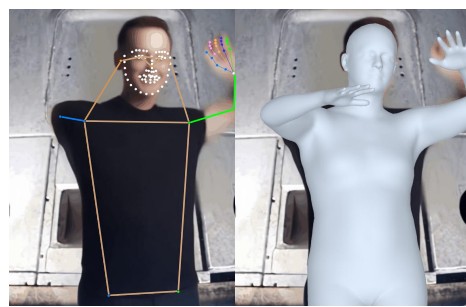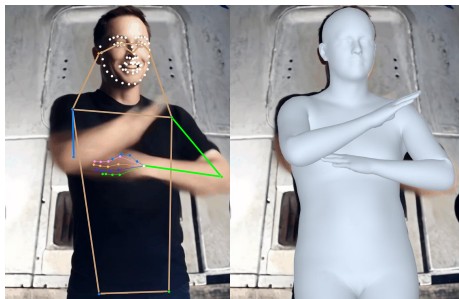

Figure 10: **The parametric 3D mesh serves as an effective object-level prior,** ensuring complete human body structures in the coarse video generated during the first stage. In the left two images, the human keypoint model fails to detect the missing right hand, which is accurately "recovered" by the parametric human mesh model. In the right two images, the human mesh model provides a more accurate prior for both blurred hands.

Table 9: **Ablation on confidence score.** We evaluate three parameter configurations corresponding to the {full motion sequence, polygon target pose, empty} conditions: (1, 0.5, 0), (0.8, 0.5, 0.2), and (3, 2, 1). The results demonstrate that performance is highly robust to the specific choice of confidence values.

| Model | Setting | SSIM | PSNR | LPIPS |
|---|---|---|---|---|
| *VividPose* | - | *0.758* | *29.83* | *0.261* |
| Ours | (1, 0.5, 0) | 0.864 | 30.08 | 0.210 |
| Ours | (0.8, 0.5, 0.2) | 0.851 | 30.10 | 0.214 |
| Ours | (3, 2, 1) | 0.873 | 30.07 | 0.217 |

videos containing incorrect human or animal structures (*i.e.*, the morphological failure rate). We evaluate 500 video pairs, each rated by three different users, resulting in 1,500 total evaluations. The results are presented in Tab. 7 of the main paper. By optimizing with a parametric 3D mesh, our approach significantly reduces incorrect human and animal structures, leading to substantial improvements in object and motion consistency.

**Parametric 3D Mesh.** Previous human image animation models mainly rely on 2D pose sequences for each frame to provide motion information. However, this approach is not optimal for general video generation. As shown in Fig. 10, we compare the results of using a parametric human mesh model (Loper et al., 2015) versus a human keypoint model (Sun et al., 2019). Our findings indicate that the human mesh model provides a robust object-level prior, which significantly benefits general video generation. Specifically, current video generation models often misinterpret the structure of humans and animals, occasionally producing unrealistic results, such as a man with three arms, an example of morphological failure. This problem becomes more serious in complex motion generation. However, incorporating human and animal priors from 3D mesh models substantially mitigates these structural inaccuracies, enabling more accurate representations of targets.

**Text Prompt and Motion Strength.** Generating videos from a single image introduces significant ambiguity. To reduce this, we incorporate additional conditioning using a text prompt and a motion strength parameter. Specifically, the text prompt defines the intended motion type, while motion strength controls the speed and complexity of motion within the video. In our experiments, we observe that varying motion strength with the same target pose leads to different motion trajectories. For instance, when moving a hand from point A to B, a video generated with low motion strength results in a direct, simple movement. In contrast, higher motion strength produces a more dynamic and complex trajectory, though it still reaches the same final pose at B. An illustrative example is provided in Fig. 11.

**Ablation on the Confidence Score for Conditioning.** We use the confidence map as simple markers to indicate whether a given spatial location contains motion supervision. This is consistent with common practice in prior work (*e.g.*, Emu Video (Girdhar et al., 2023)), where a binary mask is used purely to denote

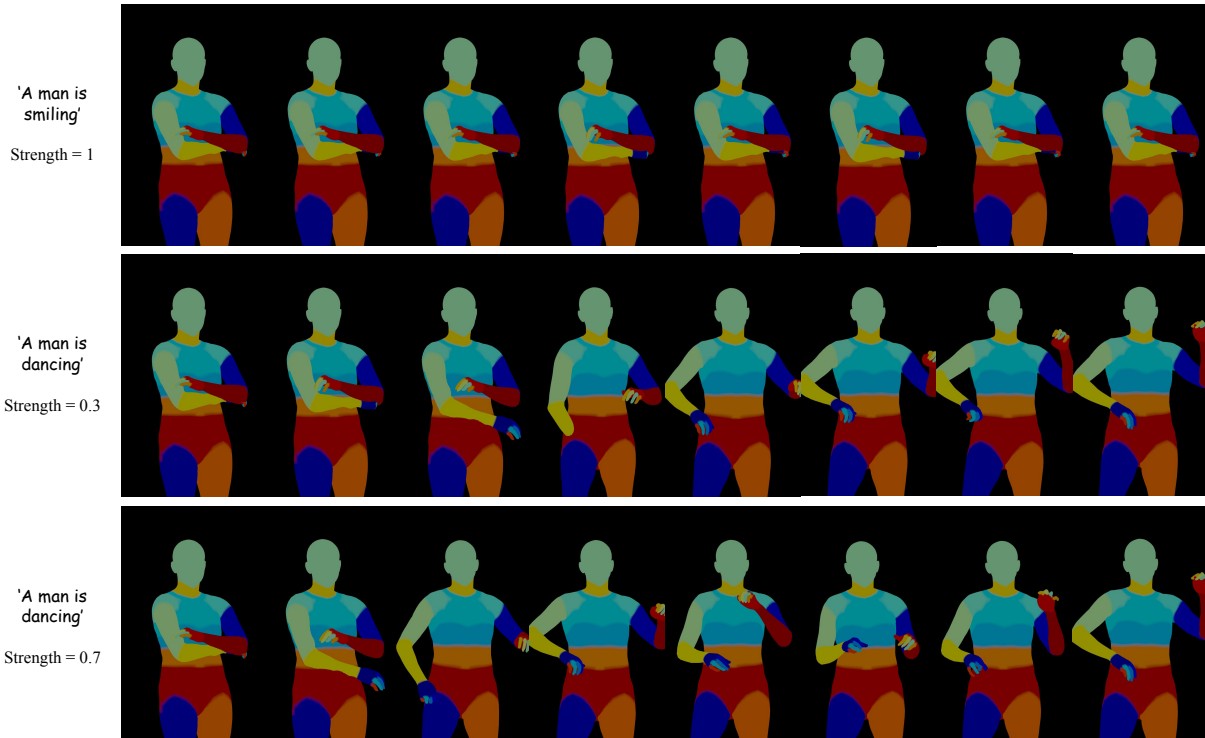

Figure 11: **Text prompt and motion strength.** We show projected motion sequences generated using different text prompts and motion strength parameters. The text prompt helps define the motion style, while the motion strength controls the speed and complexity of the generated motion. For example, a motion strength of 0.3 results in a simple, direct trajectory, whereas a strength of 0.7 produces a more dynamic and complex motion path. The same target pose is used for the second and third rows to highlight the effect of motion strength.

the presence or absence of conditioning signals. In our formulation, assigning different numeric values (*e.g.*, 1 for a full motion sequence, 0.5 for a target pose, and 0 for empty) does not introduce additional learnable meaning. Rather, these values serve as soft indicators that allow us to unify different conditioning types within a single representation, and the model does not depend on their exact magnitudes. We verify this with an ablation in Tab. 9, where we evaluate three parameter sets, *i.e.*, (1, 0.5, 0), (0.8, 0.5, 0.2), and (3, 2, 1), for {full motion sequence, polygon target pose, empty}. The results show that performance is highly robust to the specific choice of confidence values. Given this insensitivity, learning an additional confidence-prediction head is unlikely to provide meaningful gains, while it would introduce extra complexity; therefore, we adopt this simple, standard, and effective marking strategy.

## D Limitations

The proposed method has several remaining limitations. *First*, it relies on parametric 3D mesh models, requiring multiple off-the-shelf models for different object categories, though it adds only 5 seconds to the total inference time. Recent advances in 3D modeling, such as encoding 3D priors of general objects within a single diffusion model (Liu et al., 2024d), are paving the way for more general, efficient models that can be seamlessly integrated into our pipeline for high-quality video generation. *Second*, the model still struggles to generate high-quality details such as fingers and hands. *Finally*, while PMP can generate realistic motion sequences beyond 32 frames, our current implementation, based on vanilla SVD, is limited by memory constraints (80GB RAM). However, recent methods have demonstrated longer video generation ability using pretrained diffusion models (Chen et al., 2023). Exploring long-video generation with 3D knowledge remains future work.

# E  Statement of Broader Impact

The proposed ReVision has the potential to facilitate numerous fields through its advanced video generation capabilities. In the realm of creative industries, ReVision can enhance the efficiency and creativity of artists and designers by generating high-fidelity videos. The high-quality generated videos can also contribute to research on synthetic datasets by creating realistic videos, aiding in reducing the annotations required for training vision models. However, with these advancements come ethical considerations, such as the risk of generating deepfakes or other malicious content. It is thus crucial to implement safeguards to minimize potential harms.

