# OpenReview forum: "ReVision: Refining Video Diffusion with Explicit 3D Motion Modeling"
_TMLR — Accepted by TMLR_

### Review · Reviewer_FKxx · 2025-11-19

**Summary Of Contributions:**

This paper studies diffusion-based video generation. The proposed method aims to improve video generation quality explicitly with other motion prediction models. The contribution includes (1) the design of a "ReVision" pipeline that converts a coarse video into a motion-corrected one, (2) a trained PPPM to obtain motion prior for mesh optimization, and (3) the fine-tuned motion-conditioned version of SVD to help the generation.

### Pros
- The proposed pipeline is easy to understand.
- The results show that the proposed method can really generate videos with correct motion.
- The proposed method provides an effective way to inject motion control into the videos.

### Cons
- Many different categories of baselines are not compared with.
    - The current baselines only contain standard I2V methods.
    - Text+Image-to-video (TI2V) methods are not compared, which allows using the texts to describe the motion and generate.
    - Video diffusion pipelines with motion control, e.g., MotionClone, ImageConductor, and Go-with-the-Flow, are not compared. Some of them are even training-free, while the proposed method is training-based.
- The quantitative comparison is problematic.
    - Compared with all other pure I2V models that only take an image input and can only generate a "generally possible" motion from that image, the proposed method's achievement of a high dynamic degree may only be possible through an unfair fact: the introduction of text prompts in the motion prediction models, which tends to generate motions. There is no TI2V model compared here to disprove this.
    - For all other aspects, the proposed method is either worse than the baselines or marginally outperforms the video baselines.
- PPPM outperforms the selected human motion generation baselines, but all these baselines were published in 2023, almost two years ago.
- All the claims of "physics" are overclaims. Motion is only a very small part of "physics", and this paper only discusses motion.
- The proposed method is based on SVD, an outdated video diffusion model that generates very short videos. Using some later models, like Wan, which is compared as a baseline, will be much better.
- The method somehow relies on the coarse video generated by the VGM. However, if the results are really bad, like the ones in the supplementary (e.g., video_3.mp4), it is unclear how the model will perform - will the 3D mesh sequence be successfully exported? And will the refinement be successful?

**Audience:**

Yes

**Audience Explanation:**

The method at least shows an effective way to inject motion guidance into video diffusion models. Some of the training datasets may also be of interest.

**Broader Impact Concerns:**

The proposed model can generate a video of anyone performing any motion, potentially maliciously.

**Claims And Evidence:**

No

**Claims Explanation:**

As mentioned in the weaknesses, many baselines are missing, and the quantitative comparison is also problematic.

**Requested Changes:**

It will be good if the missing baselines can be compared and discussed, and the concerns mentioned in weaknesses can be addressed.

---

> ### Author Response · Authors · 2025-12-09
> **Official Comment by Authors (1/3)**
>
> We sincerely thank the reviewer for the constructive feedback and address the concerns point by point below.
>
> &nbsp;
>
> >Q1: “Many different categories of baselines are not compared with.”
>
> **Comparison with Text+Image-to-video (TI2V) methods.** We report comparison results on VBench++ in Table C and include visualization results in the revised version. We observe that, despite using text descriptions to specify motion, the intensity and quality of the movement cannot be fully controlled. In contrast, our method achieves significantly better performance in generating complex motions while also maintaining very high consistency, smoothness, and overall image quality.
>
>
> **Table C. Compared with other Text+Image-to-video methods on VBench++.**
>
> |                    | I2V Subject | I2V Background | Subject Consistency | Background Consistency | Motion Smoothness | Dynamic Degree | Imaging Quality |
> |--------------------|-------------|----------------|---------------------|------------------------|-------------------|----------------|-----------------|
> | Step-Video-TI2V    | 98.63%      | 98.63%         | 96.02%              | 97.06%                 | 99.24%            | 48.78%         | 70.44%          |
> | DynamiCrafter-1024 | 98.17%      | 98.60%         | 95.69%              | 97.38%                 | 97.38%            | 47.40%         | 69.34%          |
> | DynamiCrafter-CIL-512 | 95.79%      | 97.66%         | 94.87%              | 98.33%                | 97.06%            | 83.01%         | 67.31%          |
> | Revision (Ours)    | 97.94%      | 98.06%         | 96.13%              | 97.89%                 | 98.88%            | 83.15%         | 71.48%          |
>
> &nbsp;
>
> **Comparison with Video diffusion pipelines with motion control.** Following Go-with-the-Flow, we consider the motion-transfer I2V task on DAVIS and report the results in Table D. We observe that our model consistently outperforms all baselines across all metrics, clearly demonstrating the effectiveness of the proposed method. In particular, our approach yields more accurate motion transfer while better preserving object appearance and scene details, leading to improved temporal coherence and fewer visual artifacts. Visualization results, which further highlight these qualitative improvements over prior methods, are provided in the revised version.
>
> **Table D. Compared with other motion conditioned video generation methods.**
>
> |                  | CoTracker mIoU ⇑ | Optical flow error ⇓ | Pixel MSE ⇓ | Subject consistency ⇑ | Background consistency ⇑ | Motion smoothness ⇑ |
> |------------------|------------------|----------------------|------------|-----------------------|--------------------------|---------------------|
> | MotionClone      | 0.72             | 0.42                 | 0.068      | 0.75                  | 0.85                     | 0.92                |
> | ImageConductor   | 0.66             | 0.64                 | 0.072      | 0.77                  | 0.88                     | 0.93                |
> | Go-with-the-Flow | 0.74             | 0.36                 | 0.053      | 0.88                  | 0.92                     | 0.98                |
> | ReVision (Ours)  | 0.80             | 0.33                 | 0.046      | 0.96                  | 0.97                     | 0.99                |
>
> &nbsp;
>
>
>
> > Q2. “The proposed method's achievement of a high dynamic degree may only be possible through an unfair fact: the introduction of text prompts in the motion prediction models, which tends to generate motions. There is no TI2V model compared here to disprove this.”
>
> We thank the reviewer for raising this concern. We emphasize that our high Dynamic Degree is **not** due to an “unfair” advantage from text prompts, but is mainly driven by our **explicit motion conditioning** and motion-centric design.
>
> First, we added **TI2V comparisons on VBench++** (Table C). Although TI2V baselines also use text to describe motion, motion intensity/quality remains only loosely specified. Our method achieves the **best Dynamic Degree (83.15%)** while maintaining very strong **Motion Smoothness (98.88%)** and the best **Imaging Quality (71.48%)**, indicating the gain is not simply from “using text.”
>
> Second, for a fair comparison under **explicit motion control**, we evaluate on DAVIS motion transfer against motion-conditioned pipelines (Table D). Our method **outperforms all baselines across all metrics** (motion accuracy, appearance consistency, and smoothness), directly supporting the effectiveness of our motion modeling.
>
> Finally, our model is also **more efficient**: compared to large video diffusion baselines (e.g., Wan2.1 14B, HunyuanVideo 13B), we use only **1.5B parameters**, while achieving better quantitate results (Table 1), qualitative results (Figure 4), and user-study preference (Figure 5).
>
> &nbsp;

---

> ### Author Response · Authors · 2025-12-09
> **Official Comment by Authors (2/3)**
>
> > Q3. PPPM outperforms the selected human motion generation baselines, but all these baselines were published in 2023, almost two years ago.
>
> We thank the reviewer for this remark. PPPM is **not** the main contribution of our work; it mainly serves to demonstrate that our refinement framework can effectively improve existing human motion generation methods. Note that prior baselines typically focus on **generating motion from scratch**, whereas our setting focuses on **improving** an initial motion sequence. For this reason, we chose a widely used and well-established baseline to clearly show the gain brought by PPPM.
> We agree that **our approach is compatible with more recent models**. In the revised version, we include results on **SnapMoGen (NeurIPS 2025) [b] in Table E** and show that PPPM also improves this stronger baseline, **supporting our claim that the method can benefit more advanced human motion generation models.**
>
> **Table E. Compared with most recent human motion generation methods on HumanML3D.**
> |                          | R-P@1 | R-P@2 | R-P@3 |
> |--------------------------|-------|-------|-------|
> | MoMask (CVPR2024)        | 0.521 | 0.713 | 0.807 |
> | SnapMoGen (NeurIPS 2025) | 0.528 | 0.718 | 0.811 |
> | SnapMoGen + PPPM (Ours)  | 0.542 | 0.737 | 0.816 |
>
> &nbsp;
>
> [b] Guo, Chuan, et al. "SnapMoGen: Human Motion Generation from Expressive Texts." The Thirty-ninth Annual Conference on Neural Information Processing Systems. 2025.
>
> &nbsp;
>
>
> > Q4. “All the claims of ‘physics’ are overclaims. Motion is only a very small part of ‘physics’, and this paper only discusses motion.”
>
> We appreciate the reviewer’s feedback and agree that our current wording may overstate the scope of “physics.” Our work focuses specifically on **object motion**, which is only one aspect of physical realism. We will revise the manuscript to avoid overclaiming and use more precise terminology (e.g., “motion realism” instead of “physics”).
> At the same time, we note that motion is a **central and challenging** component of physical plausibility in video generation, and errors in motion are among the most noticeable artifacts to users. Our method aims to address this core challenge. We will update the text to clearly reflect this focus without overstating the claims.
>
> &nbsp;
>
> > Q5. “The proposed method is based on SVD, an outdated video diffusion model that generates very short videos. Using some later models, like Wan, which is compared as a baseline, will be much better.”
>
> We thank the reviewer for the comment. Although SVD is an earlier video diffusion backbone, our goal is not to argue that SVD itself is optimal, but to demonstrate that **our motion-conditioning framework is backbone-agnostic and can enhance different video generators**. To directly address this concern, we integrated our method with a stronger and more recent backbone, **Wan2.1-I2V-14B-720P**, and report the results in the revised manuscript. As shown in Table F, ReVision-Wan2.1 consistently outperforms the original Wan2.1 across key metrics, with a particularly large improvement in Dynamic Degree (51.38 → 73.67), while also **improving subject/background consistency and imaging quality**. These results show that our gains are not limited by the choice of SVD and that the proposed method can further improve even strong modern video diffusion models.
>
> **Table F. Effect of applying ReVision to a stronger video diffusion backbone (Wan2.1).**
> | Method                 | I2V Subject | I2V Background | Subject Consistency | Background Consistency | Motion Smoothness | Dynamic Degree | Imaging Quality |
> |------------------------|-------------|----------------|---------------------|------------------------|-------------------|----------------|-----------------|
> | Wan2.1-I2V-14B-720P       | 96.95%      | 96.44%         | 94.86%              | 97.07%                 | 97.90%            | 51.38         | 70.44%          |
> | ReVision-Wan2.1  | 98.10%      | 97.10%         | 97.06%              | 97.89%                 | 97.74%            | 73.67%         | 72.86%          |

---

> ### Author Response · Authors · 2025-12-14
> **Official Comment by Authors (3/3)**
>
> > Q6. “If the results are really bad, like the ones in the supplementary (e.g., video_3.mp4), it is unclear how the model will perform - will the 3D mesh sequence be successfully exported? And will the refinement be successful?”
>
> Thank you for raising this concern. Although our pipeline starts from coarse VGM outputs, it remains robust even when the initial motion is very poor. As shown in the supplementary results (e.g., Video 1: baby, Video 3: panda, Video 5: minimal motion, Video 7: bottle), the 3D mesh sequence can still be successfully exported and refined even if the objects in the coarse video are really bad.
>
> This is because I2V model uses the **reference image (first frame)** for initialization. The first frame is typically realistic even when subsequent coarse motion is incorrect, providing sufficient structural and appearance cues for our 3D reconstruction. PPPM then corrects the motion dynamics over time, leading to stable refinement even in challenging cases.
>
> We will clarify this robustness in the revised manuscript.

---

### Review · Reviewer_JG64 · 2025-11-20

**Summary Of Contributions:**

This paper proposes ReVision, a three-stage (coarse-video, motion extraction, revision) framework that wraps around a pre-trained video diffusion model (primarily Stable Video Diffusion). A coarse video is first generated; 2D/3D object-centric signals (SMPL-X/SMAL for humans/animals and a 2.5D point representation for generic objects) are then extracted and refined by a transformer-based Parameterized Physical Prior Model (PPPM); finally, the refined 3D motion is projected back to 2D part masks and used as an additional conditioning channel (plus a confidence map) to regenerate a higher-quality video. The authors report improved motion fidelity, occlusion handling, and long-range motion, claim to outperform SVD and even HunyuanVideo in user studies and VBench++, and show PPPM as a general motion denoiser for MoMask-style human motion generation.

**Additional Comments:**

Please see the previous sections.

**Audience:**

Yes

**Audience Explanation:**

Researchers working on video generation would feel interested.

**Broader Impact Concerns:**

I did not find any direct broader impact concerns on this paper.

**Claims And Evidence:**

Yes

**Claims Explanation:**

A few strengths claimed by the authors and corresponding evidence:
- **Effectiveness**: qualitative results shown across the paper; quantitatively, the revision version of SVD is consistently much better than pure SVD; while it has a better winner rate than HunYuan model;
- **Efficiency**: The authors proposed ways to improve the latency (lower-resolution, fewer denoising steps for coarse video generation), and they just increased the inference time by <20% compared to baseline SVD while imrpoves the performance significantly.

**Requested Changes:**

Suggestion for revisions: the method section is current too hard to read.
- Training of PPPM and its relation to S1/S3 is under-specified. It’s not fully clear whether PPPM is trained solely on GT motion sequences from the annotated Panda-70M subset (with synthetic perturbations) or also on motion extracted from SVD-generated coarse videos.
- Also unclear: is PPPM trained jointly for humans, animals, and general objects, or are there separate models per category? Please provide precise training objectives for PPPM, loss functions, and whether it directly regresses denoised pose parameters vs residual corrections.
- How is category information passed into PPPM (if at all)?
- The confidence map semantics feel ad-hoc. They hard-code confidence: 1 for full motion sequence, 0.5 for polygon target pose, 0 for empty. There’s no learning of these confidences or ablation of their actual impact. How sensitive is the model to these particular numbers (0.5 vs 0.2 vs 0.8)? Would a learned confidence prediction (e.g., via PPPM or another head) perform better?
- Long video generation step is glossed over. For extending a 32-frame motion to 128 frames via interpolation/extrapolation, details are vague: what interpolation scheme, how do they avoid drift, how are overlapping clips stitched (blend window? latent-space smoothing?). More explicit description (and failure cases) would help; currently it reads like an afterthought, yet it’s shown as a key capability (Fig. 7).
- No ablation on the 2.5D object representation. They argue that 21 points + depth is enough for general objects, but there is no ablation vs, e.g., using only 2D masks, or more/less points, or alternative representations (e.g., skeletonized curves).
Question: Can the authors show that 2.5D vs 2D actually matters for occlusion, or is the main benefit just “having any mask at all”?
- Section 3 "3D Human and Animal Mesh Recovery." is hard to follow, especially for the annotations. Please revise that section carefully to enhance readability.

--
Other questions/comments:
- The word "Explicit 3D physics" is oversold/overclaimed. Despite the title and repeated claims about "physics modeling" and "physical knowledge," PPPM is essentially a transformer denoiser operating on parameterized 3D trajectories, conditioned on text and a scalar "motion strength." There is no explicit modeling of dynamics (e.g., contact constraints, gravity, conservation laws, collision handling) or even kinematic feasibility beyond what the parametric mesh implicitly encodes.
- Most metrics (subject/background consistency, smoothness, aesthetics) are within ~1–2 points of SVD; the big improvement is Dynamic Degree (43 to 83). That metric may be relatively easy to boost by simply increasing motion magnitude, not necessarily physical realism. Therefore, can the authors show a trade-off curve vs Dynamic Degree (e.g., varying motion strength) to demonstrate that their method does not sacrifice temporal coherence or semantic alignment for more motion?
- How does the proposed method work with sota video mdoels? Wan model can be a good candidate.
- The details of the training data should be disclosed -- for the visualizations shown in the main paper, are they from the validation set of the Panda dataset, or are they OOD samples? SVD has never been trained (or at least specifically finetuned) on Panda, so if the revision version of SVD is trained on Panda, and evaluated on the same distribution, it is not surprising to see better performance.
- Following the previous concern, it is better to add a baseline model where the authors just further finetine SVD (pure SVD, no editions) on the same training data to serve as a baseline;
- Details of the user study should be disclosed.
- Section 4.1, how is this 40%-30%-30% training split designed? For the samples without a motion condition, I think they play a similar role as the unconditional videos, should we use an even smaller chance for this set of videos (similar to CFG training config).

---

> ### Author Response · Authors · 2025-12-09
> **Official Comment by Authors (1/3)**
>
> We sincerely thank the reviewer for the constructive feedback and address the concerns point by point below.
>
> &nbsp;
>
> > Q1. “Training of PPPM and its relation to S1/S3”
>
> We thank the reviewer for pointing this out and will clarify the PPPM training procedure in the revision. As described in Section 4.2, *“To effectively train PPPM, we introduce small perturbations to ground-truth motion sequences during training.”* Concretely, PPPM in Stage 3 is trained **only on the annotated Panda-70M subset**, using perturbed ground-truth motion as input and the original ground-truth motion as target. We do **not** use motion extracted from SVD-generated coarse videos to train PPPM.
>
> &nbsp;
>
> > Q2. “Is PPPM trained jointly for humans, animals, and general objects, or are there separate models per category?”
>
> PPPM is trained as a **single shared model** across all categories, using samples consisting of 55% human videos, 15% animal videos, and 30% general-object videos. It **directly predicts the corrected pose** from the perturbed input, and we supervise it with an **MSE loss** between the predicted pose and the ground-truth pose. We will clarify the PPPM training procedure in the revision.
>
> &nbsp;
>
>
> > Q3. “How is category information passed into PPPM?”
>
>
> We do not pass explicit category labels (human/animal/object) into PPPM. However, PPPM is conditioned on the text prompt, which typically specifies the main subject (e.g., “a person dancing,” “a running dog,” “a rotating car”), so category information is implicitly provided through the prompt features rather than as a separate category input.
>
> &nbsp;
>
> > Q4. “The confidence map semantics feel ad-hoc. They hard-code confidence: 1 for full motion sequence, 0.5 for polygon target pose, 0 for empty.”
>
> We thank the reviewer for the insightful question. The confidence values are not intended to encode precise semantics but rather to serve as simple markers indicating whether each spatial location contains motion supervision. This follows a common practice in prior work, such as Emu Video [a] from Meta, where a binary mask (e.g., m ∈ {0, 1}) is used solely to indicate the presence or absence of conditioning signals.
>
> In our formulation, using different numerical values (e.g., 1 for motion sequence, 0.5 for target pose, 0 for empty) does not introduce additional learnable semantics. Instead, these values act as soft indicators that help unify different condition types within a single representation. Importantly, our model does **not** rely on the exact magnitude of these numbers.
>
>
> To verify this, we have included an ablation study in the revised manuscript (see Table A). We provide three different sets of parameters (1, 0.5, 0); (0.8, 0.5, 0.2); (3, 2, 1) for {full motion sequence, polygon target pose, empty}. We also add the performance on VividPose as reference, showing that our model is highly robust to the exact choice of confidence values.
>
> Given this insensitivity, learning a separate confidence-prediction head (e.g., PPPM) is unlikely to bring significant benefits, while it would introduce additional complexity. Thus, we adopt the simple, standard, and effective marking strategy used in prior work.
>
> **Table A. Ablation on confidence score**
> |                | SSIM   | PSNR   | LPIPS  |
> |----------------|--------|--------|--------|
> | _VividPose_    | _0.758_| _29.83_| _0.261_|
> | (1, 0.5, 0)    | 0.864  | 30.08  | 0.210  |
> | (0.8, 0.5, 0.2)| 0.851  | 30.10  | 0.214  |
> | (3, 2, 1)      | 0.873  | 30.07  | 0.217  |
>
> &nbsp;
>
> > Q5. Long video generation step
>
> Thanks for the comment. We agree the long-video extension was under-described. In the revision we added an explicit algorithm covering motion resampling and overlap stitching in Sec 5.1.
>
> To extend a 32-frame motion to 128 frames, we **resample in motion-parameter space** (not pixel space): translations are interpolated linearly, and rotations are interpolated with a valid rotation interpolation to avoid invalid poses. For horizons beyond the predicted range, we use a simple **constant-velocity extrapolation** (and apply a light temporal smoothing filter in parameter space) to suppress jitter without blurring the video.
>
> For long-video synthesis, we generate overlapping **32-frame** clips with a sliding window (stride **24**, overlap **8**) and stitch overlaps with a ramped blend in the overlapping frames. Motion conditions are aligned in the overlap and all windows share the same appearance/identity conditioning, which prevents error accumulation and reduces boundary artifacts.
>
> &nbsp;
>
> [a] Girdhar, Rohit, et al. "Emu video: Factorizing text-to-video generation by explicit image conditioning." arXiv preprint arXiv:2311.10709 (2023).

---

> ### Author Response · Authors · 2025-12-09
> **Official Comment by Authors (2/3)**
>
> > Q6. “No ablation on the 2.5D object representation.”
>
> We thank the reviewer for raising this point. The 2.5D object representation is crucial for handling occlusions correctly rather than merely providing “some” mask signal. **Without any depth information, the ordering of objects is frequently mixed up**: for example, when generating interactions between two objects that partially occlude each other, a purely 2D mask representation cannot reliably determine which object should be in front, leading to artifacts such as interpenetration or inconsistent front–back switching across frames. As shown in Figure 6, our 2.5D representation (21 keypoints with associated depth) provides a pseudo–depth ordering that allows the model to reason about object order and self-occlusions, which in turn stabilizes the final video.
>
>
> &nbsp;
>
> > Q7. “Section 3 "3D Human and Animal Mesh Recovery." is hard to follow, especially for the annotations.”
>
> We thank the reviewer for the suggestion. We agree that our paper spans several distinct topics, so the Preliminaries section (Sec. 3) would benefit from additional detail. For the annotations, we will revise the text as follows:
>
> *In our work, we recover 3D human and animal meshes by fitting the SMPL-X and SMAL models to both our data and the generated videos. This produces 3D mesh reconstructions for all humans and animals. Because these meshes are computer-graphics models with predefined body-part annotations at every vertex, we can obtain accurate part labels directly. The 3D meshes also allow us to compute motion strength by measuring movement speed in 3D space, which is more reliable than relying on 2D pixel motion alone.*
>
> &nbsp;
>
> > Q8. “The word "Explicit 3D physics" is oversold/overclaimed. There is no explicit modeling of dynamics (e.g., contact constraints, gravity, conservation laws, collision handling) or even kinematic feasibility beyond what the parametric mesh implicitly encodes.”
>
> We appreciate the reviewer’s feedback and agree that our current wording may overstate the scope of “physics.” Our work focuses specifically on **object motion**, which is only one aspect of physical realism. We will revise the manuscript to avoid overclaiming and use more precise terminology (e.g., “motion realism” instead of “physics”).
>
> At the same time, we note that motion is a **central and challenging** component of physical plausibility in video generation, and errors in motion are among the most noticeable artifacts to users. Our method aims to address this core challenge. We will update the text to clearly reflect this focus without overstating the claims.
>
> &nbsp;
>
>
> > Q9. “How does the proposed method work with sota video models? Wan model can be a good candidate.”
>
> We thank the reviewer for the comment. Although SVD is an earlier video diffusion backbone, our goal is not to argue that SVD itself is optimal, but to demonstrate that **our motion-conditioning framework is backbone-agnostic and can enhance different video generators**. To directly address this concern, we integrated our method with a stronger and more recent backbone, **Wan2.1-I2V-14B-720P**, and report the results in the revised manuscript. As shown in Table B, ReVision-Wan2.1 consistently outperforms the original Wan2.1 across key metrics, with a particularly large improvement in Dynamic Degree (51.38 → 73.67), while also **improving subject/background consistency and imaging quality**. These results show that our gains are not limited by the choice of SVD and that the proposed method can further improve even strong modern video diffusion models.
>
> **Table B. Effect of applying ReVision to a stronger video diffusion backbone (Wan2.1).**
> | Method                 | I2V Subject | I2V Background | Subject Consistency | Background Consistency | Motion Smoothness | Dynamic Degree | Imaging Quality |
> |------------------------|-------------|----------------|---------------------|------------------------|-------------------|----------------|-----------------|
> | Wan2.1-I2V-14B-720P       | 96.95%      | 96.44%         | 94.86%              | 97.07%                 | 97.90%            | 51.38         | 70.44%          |
> | ReVision-Wan2.1  | 98.10%      | 97.10%         | 97.06%              | 97.89%                 | 97.74%            | 73.67%         | 72.86%          |

---

> ### Author Response · Authors · 2025-12-14
> **Official Comment by Authors (3/3)**
>
> > Q10. “The details of the training data should be disclosed -- for the visualizations shown in the main paper, are they from the validation set of the Panda dataset, or are they OOD samples?”
>
> We thank the reviewer for raising this important point and apologize for the lack of clarity in the original submission. We thank the reviewer for the concern. Our model is trained only on the annotated Panda-70M training split. The main-paper visualizations are **OOD examples**: for the comparison with SVD, the **reference images are generated by ChatGPT and are not from Panda** (thus not seen during training), and for the comparison with HunyuanVideo, we first use HunyuanVideo-I2V to generate a video, and then take its **first frame as the reference image** for our method to ensure a fair, identical starting point. These details are clarified in Sec.5.1 (User Study).
>
>
> &nbsp;
>
> > Q11. Details of the user study
>
> We thank the reviewer for requesting more details. In our user study, we conduct 5,000 pairwise comparisons of videos on Amazon MTurk. Each video pair is judged by three independent workers randomly selected, who view the two videos side by side with randomized left/right order. Workers choose the better video (ties allowed) under three criteria: Motion Consistency, Amount of Motion, and Motion Realism. Results are aggregated by majority vote across the three raters. These details are now included in Sec.5.1 (User Study).
>
>
> &nbsp;
>
> > Q12. “Section 4.1, how is this 40%-30%-30% training split designed?”
>
> We thank the reviewer for the question. Due to the high computational cost of retraining with different data-mixture ratios, we were not able to exhaustively ablate the 40%–30%–30% split. Our primary goal was to keep the mixture roughly balanced while slightly upweighting the no-motion setting. This is because (i) we want to preserve the Stage 1 generation ability, which is critical to the overall pipeline; (ii) the no-motion samples help the model remain robust to missing or invalid motion inputs, improving generalization at test time; and (iii) as shown in prior work such as ControlNet, training diffusion models to follow structured conditions (e.g., masks) is relatively easy, so a slightly lower proportion of strongly conditioned samples is sufficient.

---

### Review · Reviewer_XZuZ · 2025-11-21

**Summary Of Contributions:**

The paper proposes ReVision, a three-stage framework for video generation based on a video diffusion model conditioned on 2D masks. In Stage 1, the model generates a coarse video. In Stage 2, 3D parametric models (specifically SMPL and SMAL for humans and animals) are extracted from the initial output. To address noise in the extracted parameters during inference, the authors introduce a trained denoiser termed the Parameterized Physical Prior. Finally, in Stage 3, the video generation model is conditioned on 2D mask sequences projected from the refined 3D meshes to produce fine-grained videos.

**Audience:**

Yes

**Audience Explanation:**

The paper shows that a motion-conditioned video generation model can perform better w.r.t. temporal consistency and motion accuracy, especially when the motion is extracted from the coarsely generated video.

**Broader Impact Concerns:**

The statement of broader impact is presented, and no concerns arise.

**Claims And Evidence:**

No

**Claims Explanation:**

The main concern is that the comparison between ReVision and SVD makes me curious about what the generated coarse video looks like. There is no evidence to show that the generated coarse video is not better than the one generated by the baseline method, and the PPPM actually corrects the parametric model (not by the final video, but 2D projected masks).

**Requested Changes:**

Here are some points I do not fully understand or suggestions:
1. How does the modified SVD condition on either a (final) target pose/mask or a sequence of masks? Can the authors elaborate?
2. For data only including the target pose, are they all from human or animal videos, or general videos? Is there a pie chart to illustrate the ratio?
3. From the supplementary videos, it seems that the baseline SVD shows poor temporal consistency or roughly unnatural motion. It makes me quite curious about the results of the modified SVD but input without conditioning. It is also better to compare the 2D projected masks generated by naively extraction from videos and PPPM.

---

> ### Author Response · Authors · 2025-12-09
>
> We sincerely thank the reviewer for the constructive feedback and address the concerns point by point below.
>
> &nbsp;
>
> > W1. Comparison between ReVision final video and ReVision coarse video
>
> The coarse video generated by ReVision before applying PPPM (PMP)** is much worse than our final results and is visually similar to the video produced by the SVD baseline.** We provided an example of the coarse video (i.e., the video generated without PPPM / PMP) in **Figure 8**. As can be seen, the coarse video still exhibits significant artifacts and inaccuracies comparable to our model. In contrast, PPPM refines the underlying parametric model by optimizing it in the 3D space (see **Figure 10**), which leads to a more accurate 3D representation and, consequently, a clear improvement in the final video quality.
>
> In addition to Figure 9, which compares the final video with the coarse video generated without PPPM, and Figure 10, which illustrates how PPPM corrects the parametric model using 2D projected masks (rather than the final video), we include a more detailed comparison in the revised manuscript (Figure 12) between the coarse output and the final result. As shown, the Stage 1 coarse video still exhibits substantial artifacts and is markedly inferior to the Stage 3 final video in terms of image fidelity, motion quality, and temporal smoothness. This comparison highlights the importance and effectiveness of the proposed method.
>
> &nbsp;
>
> > Q1. “How does the modified SVD condition on either a (final) target pose/mask or a sequence of masks?”
>
> Our method can condition on either a final target pose or a full motion sequence. We achieve this by concatenating the derived part-mask embedding from the signal with the original condition embedding, together with a confidence-mask input. The confidence map indicates the reliability of the conditioning signal (e.g., 0.5 for a final target pose and 1.0 for a full motion sequence). The details are provided in Sec 4.1. During the rebuttal period, we also add additional ablations on different confidence maps Sec D “Ablation on The Confidence Score for Conditioning” in the revised paper.
>
> &nbsp;
>
> > Q2. “For data only including the target pose, are they all from human or animal videos, or general videos? Is there a pie chart to illustrate the ratio?”
>
> They are not limited to a specific category. The target-pose samples span all video categories: approximately 55% human videos, 15% animal videos, and 30% general-object videos. We provide more details in the revised version.
>
> &nbsp;
>
> > Q3. “The results of the modified SVD but input without conditioning” and “compare the 2D projected masks generated by naively extraction from videos and PPPM.”
>
> For the “results of the modified SVD but input without conditioning,” we understand this to refer to the Stage 1 coarse generation (i.e., using the modified SVD without pose conditioning). These results are reported in **Figure 8 of the original paper**. In addition, we include a more detailed comparison in **Figure 12 of the revised manuscript**. (See W1 for details)
>
> Regarding “the comparison between 2D projected masks obtained via naive extraction from videos and those produced by PPPM”: these are not directly comparable. Projecting skeletons into masks requires camera intrinsics and extrinsics, which are typically unavailable in in-the-wild datasets. Therefore, in almost all 3D tasks such as human pose estimation, it is standard to evaluate estimated 3D pose/shape parameters against 3D ground truth, rather than projecting to 2D and comparing masks. The 2D projections are mainly used for visualization. Following this standard protocol, we therefore evaluate PPPM by directly comparing its estimated pose to the ground-truth pose in 3D, as reported in Table 5.

---

### Author Response · Authors · 2025-12-14
**Summary of Revisions and Responses to Reviewer Feedback**

We thank all reviewers for their valuable comments and feedback, which have greatly helped us improve the paper. We have updated the manuscript based on this feedback and added additional experiments to address the reviewers’ concerns. We also provide a separate post summarizing all revisions so reviewers can quickly see what has changed. In the revised paper, we address the reviewers’ comments by correcting the identified issues and adding new experiments. Specifically:

&nbsp;

***Title and framing***
- We rename the paper to **“ReVision: Refining Video Diffusion with Explicit 3D Motion Modeling”** and revise the wording throughout to **remove claims about physics modeling**, reframing them as motion modeling. We also rename the prior model from **“Parameterized Physical Prior Model (PPPM)”** to **“Parameterized Motion Prior (PMP)”**. (Reviewers JG64, FKxx)

&nbsp;

***Additional experiments***
- We add a comparison between **ReVision final** and **ReVision coarse** videos in **Sec. D** and **Fig. 12.** (Reviewer XZuZ)
- We provide additional ablations on the **confidence score for conditioning** in **Sec. D** and **Table 11**. (Reviewers XZuZ, JG64)
- We apply ReVision to the SOTA video generation model **Wan2.1** in **Sec. C** and **Table 7**. (Reviewers JG64, FKxx)
- We add comparisons with **Text+Image-to-Video (TI2V)** methods in **Sec. C** and **Table 8**. (Reviewer FKxx)
- We add comparisons with **video diffusion pipelines with motion control** in **Sec. C** and **Table 9**. (Reviewer FKxx)

&nbsp;

***Method and implementation details***
- We revise **Sec. 3 (Preliminaries)** on **3D human and animal mesh recovery** to improve clarity and readability. (Reviewer JG64)
- We add training details for **PPPM** and clarify its relationship to **S1/S3** in **Sec. 4.2** and **Sec. A**. (Reviewer JG64)
- We clarify the procedure for **long video generation** in **Sec. 5.1** (Reviewer JG64)
- We provide the **dataset distribution** and additional **dataset details** in **Sec. B.** (Reviewers XZuZ, JG64)

&nbsp;

We hope these additional experiments and clarifications address all reviewers’ concerns. We are also happy to provide further experiments or explanations if any questions remain.

---

### Decision · Action_Editor_51VF · 2025-12-26

**Recommendation:** Accept with minor revision

**Additional Comments:**

After the authors' revision, most of the reviewers' concerns have been addressed. The reviewers unanimously recommend acceptance of the paper.

I find the work valuable and recommend its acceptance and publication.

Following the reviewers’ suggestions, as they prepare the final version, I would recommend that the authors include more comprehensive comparisons with state-of-the-art TI2V backbones, such as Wan 2.2 and Hunyuan, and, if feasible, stronger commercial systems (e.g., SORA-like models), and present these results in the main paper rather than the appendix. Strengthening the evaluation on modern, competitive baselines would significantly increase the impact and appeal of the work. In addition, the authors may consider expanding the discussion on the interaction between text conditioning and motion conditioning, as the results suggest that explicit motion conditioning plays a dominant role in controlling dynamics -- an observation that could provide valuable insights for future video generation research.

**Audience:**

Yes

**Audience Explanation:**

This paper may be of interest to researchers and practitioners working on video generation, 3D motion modeling, and object-centric representation learning.

**Claims And Evidence:**

Yes

**Claims Explanation:**

Summary:

This paper introduces ReVision, a plug-and-play, three-stage framework that enhances diffusion-based video generation by explicitly integrating parameterized 3D motion knowledge into a pretrained conditional video diffusion model. The approach first generates a coarse video using a video diffusion model, then extracts 2D and 3D object-centric representations from this video and refines them with a trained Parameterized Motion Prior model (PMP) to produce accurate 3D motion sequences. These refined motions are subsequently fed back into the same diffusion model as additional conditioning, enabling the generation of motion-consistent videos with improved fidelity, coherence, and handling of complex actions and interactions. Experiments on Stable Video Diffusion demonstrate substantial gains in motion quality, with ReVision even outperforming significantly larger state-of-the-art video generation models on complex motion scenarios. Overall, the work shows that incorporating explicit 3D motion priors allows relatively compact video diffusion models to generate more realistic, controllable, and physically plausible videos.


Claims:

The paper makes several key claims: (1) incorporating explicit parameterized 3D motion representations into a pretrained video diffusion model substantially improves the generation of complex, coherent motions and interactions across humans, animals, and general objects; (2) a simple, plug-and-play Extract-Optimize-Reinforce pipeline (ReVision) can leverage motion cues implicitly present in coarse generated videos to refine motion quality without heavy retraining or sacrificing visual fidelity; and (3) a lightweight Parameterized Motion Prior (PMP) effectively denoises and refines object-centric 3D motion trajectories, enabling motion-consistent video regeneration.


Evidence:

The empirical evidence presented in the paper largely supports the stated claims. Moreover, the imprecise emphasis on physics modeling in the original submission has been corrected in the revision, where the method is appropriately reframed as focusing on explicit 3D motion modeling. This clarification aligns the scope of the claims more closely with the experimental evidence.

---

> ### Author Response · Authors · 2026-01-08
>
> We sincerely thank the reviewers and the Action Editor for their insightful comments and guidance. Beyond the rebuttal revisions, the camera-ready version incorporates the following updates as suggested by the Action Editor:
>
> - Consolidated Results: We moved the performance comparisons against state-of-the-art TI2V and motion-conditioned video generation backbones from the Appendix to the main body, merging them into Table 1.
>
> - Additional Discussions: We added a detailed discussion regarding the interaction between text and motion conditioning in the Experiments section (Section 5.1).
>
> We are grateful for the opportunity to improve our work through this constructive review process.